# Defective RNA polymerase III is negatively regulated by the SUMO-Ubiquitin-Cdc48 pathway

Zheng Wang[1]*, Catherine Wu[1], Aaron Aslanian[1,2†], John R Yates III[2], Tony Hunter[1]*

[1]Molecular and Cell Biology Laboratory, Salk Institute for Biological Studies, La Jolla, United States; [2]The Scripps Research Institute, La Jolla, United States

**Abstract** Transcription by RNA polymerase III (Pol III) is an essential cellular process, and mutations in Pol III can cause neurodegenerative disease in humans. However, in contrast to Pol II transcription, which has been extensively studied, the knowledge of how Pol III is regulated is very limited. We report here that in budding yeast, *Saccharomyces cerevisiae*, Pol III is negatively regulated by the Small Ubiquitin-like MOdifier (SUMO), an essential post-translational modification pathway. Besides sumoylation, Pol III is also targeted by ubiquitylation and the Cdc48/p97 segregase; these three processes likely act in a sequential manner and eventually lead to proteasomal degradation of Pol III subunits, thereby repressing Pol III transcription. This study not only uncovered a regulatory mechanism for Pol III, but also suggests that the SUMO and ubiquitin modification pathways and the Cdc48/p97 segregase can be potential therapeutic targets for Pol III-related human diseases.

DOI: https://doi.org/10.7554/eLife.35447.001

*For correspondence:
zhengwang.zwang@gmail.com
(ZW);
hunter@salk.edu (TH)

Present address: †Illumina, San Diego, United states

## Introduction

Eukaryotes have three conserved DNA-directed RNA polymerases (RNA Pols) (*Roeder and Rutter, 1969, 1970*; *Weinmann and Roeder, 1974*; *Zylber and Penman, 1971*), where Pol I transcribes most of the rRNAs, Pol II transcribes mRNA, and Pol III transcribes tRNA, 5S rRNA, as well as some non-coding RNAs, such as the U6 snRNA involved in mRNA splicing. The Pol III machinery includes the polymerase itself (composed of 17 subunits), as well as basal transcription factors TFIIIA, the TFIIIB complex, and the TFIIIC complex (*Geiduschek and Kassavetis, 2001*). In budding yeast, *Saccharomyces cerevisiae*, TFIIIB is composed of Brf1, Bdp1, and TBP. TFIIIC is composed of Tfc1, Tfc3, Tfc4, Tfc6, Tfc7, and Tfc8. For 5S rRNA transcription, all three basal transcription factors are required, whereas tRNA transcription only requires TFIIIB and TFIIIC. As important as it is for normal cell physiology, Pol III plays critical roles in pathological processes, such as virus infection (*Chiu et al., 2009*), tumorigenesis (*White, 2004*), and aging (*Filer et al., 2017*). In addition, Pol III mutations were recently found to cause neurodegenerative diseases in humans. Mutations that cause hypomyelinating leukodystrophy with 4H syndrome occur predominantly in the largest two subunits of Pol III, POLR3A and POLR3B (Rpc160 and Rpc128 in yeast, respectively) (*Bernard et al., 2011*; *Saitsu et al., 2011*; *Shimojima et al., 2014*; *Synofzik et al., 2013*; *Terao et al., 2012*; *Tétreault et al., 2011*), with a few in POLR1C (Rpc40 in yeast) (*Thiffault et al., 2015*), a subunit shared by Pol I and Pol III. Four mutations in BRF1 were found to cause a cerebellar-facial-dental syndrome (*Borck et al., 2015*).

How Pol III transcription is regulated is still poorly understood. Current knowledge of Pol III regulation is largely limited to phosphorylation of the Pol III machinery components, such as Maf1 (*Moir et al., 2006*; *Oficjalska-Pham et al., 2006*; *Roberts et al., 2006*) and the Rpc53 subunit of Pol

III (*Lee et al., 2012*). Maf1 is a robust Pol III repressor (*Boguta, 2013*; *Moir and Willis, 2013*). Upon stress, Maf1 is dephosphorylated and translocated into the nucleus, where it binds Pol III and blocks its interaction with TFIIIB (*Desai et al., 2005*; *Moir et al., 2006*; *Roberts et al., 2006*; *Vannini et al., 2010*). Phosphorylation of Rpc53 by the Mck1 and Kns1 kinases also represses Pol III under stress conditions, although the mechanism is unclear (*Lee et al., 2012*). SUMO is another potential regulator for Pol III that could act as a transcriptional repressor (*Neyret-Kahn et al., 2013*; *Rohira et al., 2013*), or activator (*Chymkowitch et al., 2017*), but how sumoylation regulates Pol III is largely unclear. Therefore, deeper insights regarding the regulatory mechanisms of Pol III transcription are needed to design therapeutic tools that can be used to modulate Pol III activity accordingly in human diseases.

Post-translational modification by SUMO is a conserved pathway and is essential for viability in most organisms (*Kerscher et al., 2006*). Similar to ubiquitin modification, SUMO is conjugated to a lysine residue within the target protein through a cascade of reactions catalyzed by a SUMO-specific E1 activating enzyme (*Johnson et al., 1997*), an E2 conjugating enzyme (*Johnson and Blobel, 1997*), and E3 ligases (*Johnson and Gupta, 2001*; *Strunnikov et al., 2001*; *Takahashi et al., 2001*; *Zhao et al., 2004*). SUMO proteases are responsible for both the maturation of the SUMO polypeptide (*Li and Hochstrasser, 1999*) and the removal of SUMO from modified proteins (*Li and Hochstrasser, 2000*). Sumoylation can trigger ubiquitylation through the activity of SUMO-Targeted Ubiquitin E3 Ligases (STUbLs) (*Mullen and Brill, 2008*; *Prudden et al., 2007*; *Sun et al., 2007*; *Xie et al., 2007*). Besides ubiquitylation, SUMO can also recruit the Cdc48 (p97)-Ufd1-Npl4 segregase complex, through the SUMO-interacting motifs (SIMs) in Cdc48 and Ufd1 (*Bergink et al., 2013*; *Nie et al., 2012*). The key to fully understand the functions of sumoylation is its substrates. Ever since its discovery two decades ago (*Mahajan et al., 1997*; *Matunis et al., 1996*; *Okura et al., 1996*), biochemical approaches have been greatly improved to identify thousands of sumoylated proteins, as well as their conjugation sites (*Hendriks et al., 2014*; *Lamoliatte et al., 2014*; *Tammsalu et al., 2014*), underscoring the importance of this modification in the cell. However, how sumoylation affects the functions of its protein substrates is still a challenging question that remains largely unanswered, because mutating the conjugation sites usually does not cause any obvious phenotype. To address this issue, a phenotype-based genetic method is needed.

## Results

### A reverse suppressor screen identified Pol III as a major functional target of SUMO

We designed a reverse suppressor screen in budding yeast, *Saccharomyces cerevisiae*, with the goal of identifying proteins or pathways, for which loss of sumoylation results in a phenotype. Specifically, the screen looks for lethal or sick mutations that can be rescued by a dominant Q56K mutation in SUMO (*SMT3-Q56K*). *SMT3-Q56K* is one of the SUMO pathway mutations identified previously in the *mot1-301* suppressor screen (*Wang et al., 2006*) (*Figure 1—figure supplement 1*), and it suppresses *mot1-301* dominantly (*Figure 1A, B*). *SMT3-Q56K* cells are viable, suggesting the mutated protein is partially functional (data not shown). To perform the screen (*Figure 1C*, Materials and methods), yeast cells were first transformed with a plasmid carrying *URA3* and *SMT3-Q56K*, followed by random mutagenesis, and allowed to grow into single colonies. Using the *ade2/ade3* color assay, yeast cells will turn red in the presence of the plasmid. If a clone carries a lethal/sick mutation that can be rescued by *SMT3-Q56K*, the cells can no longer lose the plasmid in order to grow. All cells from this clone will thus maintain the plasmid, forming a colony that is uniformly red. Such colonies will be sensitive to 5-fluoroorotic acid (5FOA), which counter-selects the *URA3* gene on the plasmid. Mutated genes can subsequently be cloned by transforming with a genomic DNA library and screening for 5FOA-resistant colonies. Mutations are then identified by PCR sequencing of the gene locus.

The screen results are summarized in *Table 1*. First, the screen identified mutations in expected genes, including *MOT1* and *SMT3*. The screen also revealed mutations in *AOS1* (SUMO E1) and *ULP2* (SUMO protease), which was not surprising, as they encode enzymes in the SUMO pathway. Strikingly, the remaining 13 mutations were all in genes encoding components of the Pol III transcription machinery, including the largest two subunits of Pol III (Rpc160 and Rpc128), a TFIIIB subunit (Brf1), and two TFIIIC subunits (Tfc1 and Tfc6). To confirm the screen results, the identified

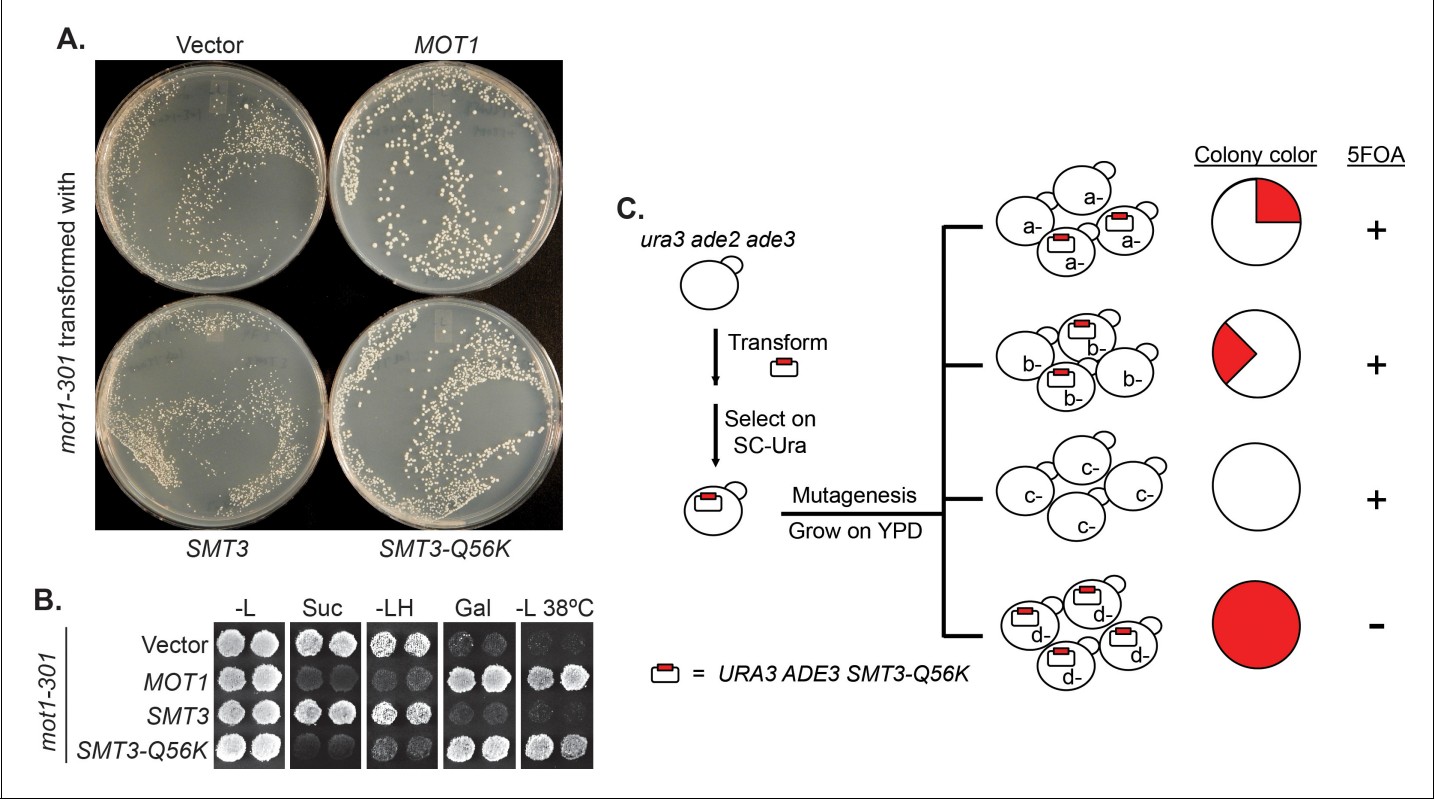

**Figure 1.** A reverse suppressor screen using the dominant *SMT3-Q56K* mutant. (**A**) A *mot1-301* strain was transformed with *CEN LEU2* vectors carrying indicated genes, then selected for transformants on SC-Leu plates. Wild-type *MOT1* or *SMT3-Q56K* made *mot1-301* cells grow faster. (**B**) Transformants from (**A**) were patched on SC-Leu (-L) then replica plated to SC-Leu-His (-LH), YPSucrose (Suc), SC-Galactose (Gal) plates, or a SC-Leu plate incubated at elevated temperature 38°C (-L 38°C). *mot1-301* is His⁺ (Spt⁻), Suc⁺ (Bur⁻), Gal⁻, and Ts⁻, whereas *SMT3-Q56K* reversed all four phenotype. The suppression is dominant because the wild-type genomic copy of *SMT3* was present in all the strains. (**C**) Design of the screen. The starting strain is an *ura3 ade2 ade3* triple mutant. *ura3* is used for *URA3* plasmid selection and 5FOA-sensitivity test. *ade2 ade3* double mutant colonies are white, but the wild-type *ADE3* on the plasmid complements *ade3* and turns the cells red. A mutant (d-) that requires the plasmid for viability will form a uniformly red and 5FOA-sensitive colony.

DOI: https://doi.org/10.7554/eLife.35447.002

The following source data and figure supplements are available for figure 1:

**Figure supplement 1.** *mot1-301* suppressor mutations.

DOI: https://doi.org/10.7554/eLife.35447.003

**Figure supplement 1—source data 1.** Source data for *Figure 1—figure supplement 1*

DOI: https://doi.org/10.7554/eLife.35447.004

mutations were introduced into wild-type cells, and subsequently crossed with a strain lacking a major SUMO E3 ligase, Siz1 (*siz1Δ*). For example, introduction of the M809I mutation in Rpc160 (*Figure 2A*) and the A704T mutation in Rpc128 (*Figure 2B*) caused severe growth defects, while *siz1Δ* fully rescued *rpc160-M809I* and partially rescued *rpc128-A704T*, as expected. However, the deletion of the closely related SUMO E3 ligase, *SIZ2*, did not rescue (*Figure 2C*). Correlating with the growth phenotype, the amount of total tRNA (*Figure 2D*) as well as individual tRNA species, including mature and pre-mature intron-containing tRNAs (*Figure 2E*), were dramatically decreased in *rpc160* mutant cells, but were restored to normal levels by *siz1Δ*. Interestingly, *siz1Δ* did not further increase tRNA levels in wild-type *RPC160* cells. No change in 5S rRNA was observed, which is a common phenomenon. This is likely because 5S rRNA is produced in excess, and that Pol III has a higher affinity for the initiation complex containing TFIIIA, which is required for the transcription of 5S rRNA, but not tRNAs (*Stettler et al., 1992*).

Based on the Cryo-EM structure of Pol III (*Hoffmann et al., 2015*), most of the mutations identified by the screen occurred on residues close to the bound DNA template or the growing RNA chain

Research article

**Table 1.** Summary of mutations rescued by *SMT3-Q56K*.

| Gene | Protein | # of alleles | Mutations |
|------|---------|--------------|-----------|
| MOT1 | Negative regulator of TBP | 3 | mot1-399 (Q1587 Stop) |
| | | | mot1-517 (G1410R) |
| | | | mot1-753 (G1300S) |
| SMT3 | SUMO | 4 | Not sequenced |
| AOS1 | SUMO E1 | 1 | aos1-492 (G56S) |
| ULP2 | SUMO protease | 4 | ulp2-4 (S108 Stop) |
| | | | ulp2-253 (G265D) |
| | | | ulp2-527 (W532 Stop) |
| | | | ulp2-63 (W532 Stop) |
| RPC160 | RNA Pol III subunit | 8 | rpc160-58 (M809I) |
| | | | rpc160-85 (G1297D) |
| | | | rpc160-33 (T379I) |
| | | | rpc160-419 (A880T) |
| | | | rpc160-426 (E282K) |
| | | | rpc160-480 (G1098D) |
| | | | rpc160-628 (R365K) |
| | | | rpc160-211 (G606S) |
| RPC128 | RNA Pol III subunit | 2 | rpc128-202 (A704T) |
| | | | rpc128-578 (D501N) |
| BRF1 | TFIIIB subunit | 1 | brf1-137 (S271L) |
| TFC1 | TFIIIC subunit | 1 | tfc1-321 (N255K, Fs) |
| | | | (AAC-AAAC, Ins, Fs) |
| TFC6 | TFIIIC subunit | 1 | tfc6-192 (G391E) |

DOI: https://doi.org/10.7554/eLife.35447.005

(*Figure 2—figure supplement 1*), suggesting that the mutations could impact Pol III enzyme activity and thus cause severe growth defects. However, none of these mutants have been reported before, so their actual enzymatic defects are unclear. We therefore tested whether previously described Pol III mutations, including *rpc31-236* which is an initiation-defective mutant (*Thuillier et al., 1995*), as well as two elongation mutants, *rpc160-112* (*Dieci et al., 1995*) and *rpc160-270* (*Thuillier et al., 1996*), could be rescued by reduced sumoylation, and found that all three mutants were rescued by *siz1Δ* (*Figure 2—figure supplement 2A*). Besides these loss-of-function mutations, when expression of wild type *RPC160* was reduced by growth of a strain in which the only *RPC160* gene was under the *GAL1* promoter in glucose, the resultant slow growth phenotype was partially rescued by *siz1Δ* (*Figure 2—figure supplement 2B*). We also tested whether human Pol III mutations that cause neuronal diseases, which were introduced into yeast Rpc160, Rpc128, and Brf1 at corresponding positions based on sequence homology, could be rescued when sumoylation was compromised. Among the 17 Rpc160 mutations tested for growth under normal conditions and at elevated temperature (37°C) (*Figure 2—figure supplement 3*), two single (Q608K and E1329K) and two double mutations (D384N, N789I and Q608K, G1308S) caused growth defects, which could all be rescued by *siz1Δ*, except for E1329K (*Figure 2—figure supplement 2C*). For Rpc128, only one of the five single mutations (L1027P) showed slower growth, which was rescued by *siz1Δ* (*Figure 2—figure supplement 2D*). All *rpc128* double mutations were lethal, and not rescued by *siz1Δ* (*Figure 2—figure supplement 3*). For the four *brf1* single mutations, three showed growth defects, two of which were rescued by *siz1Δ* (*Figure 2—figure supplement 2E*). These results confirmed the roles of SUMO in Pol III transcription, suggesting that SUMO can repress Pol III but the effect is most obvious when Pol III activity is greatly reduced either through decreased expression or inactivating mutation.

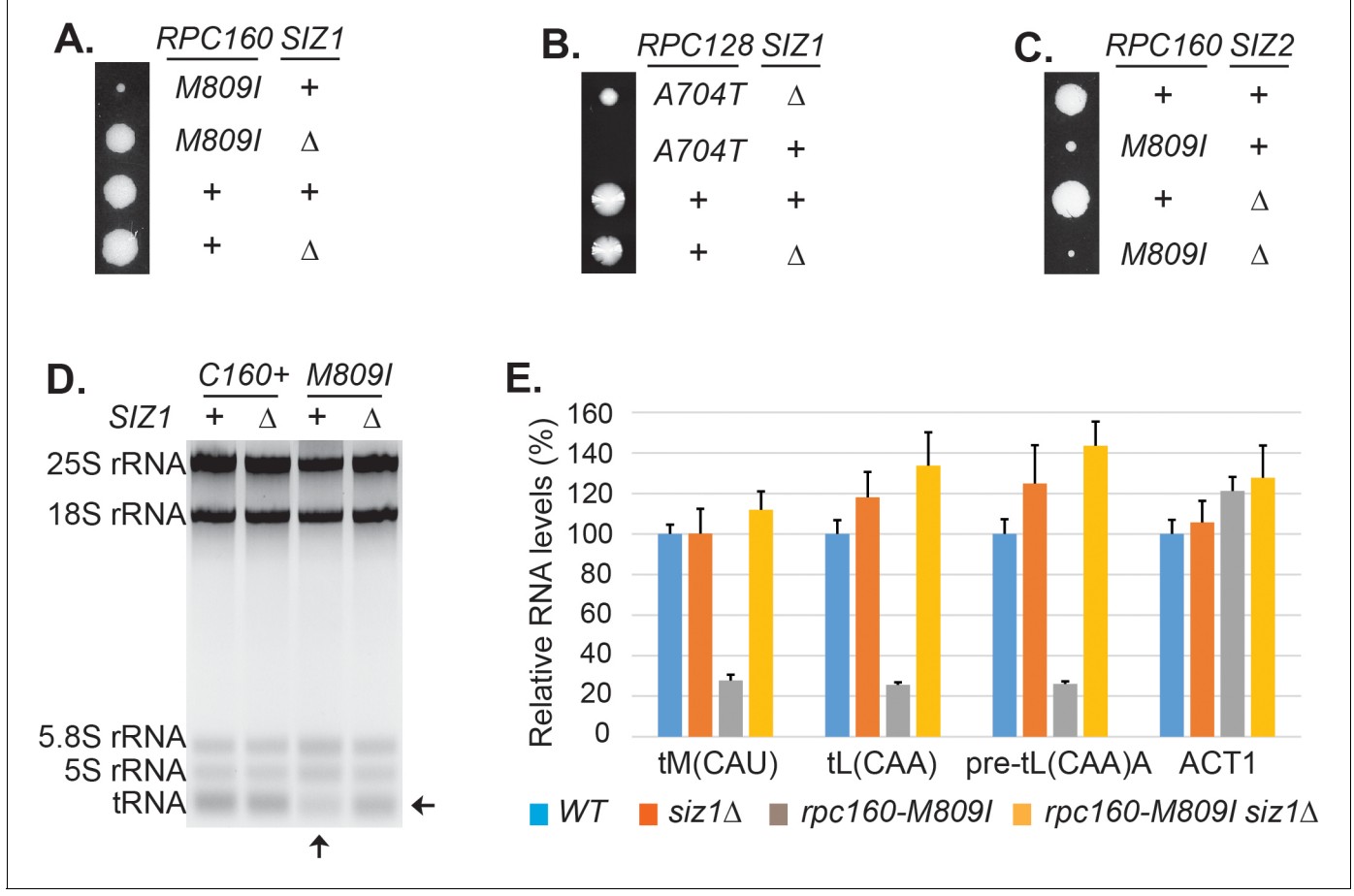

**Figure 2.** Disrupting sumoylation rescues Pol III mutations. (**A**) Tetrad analysis of a cross between *rpc160-M809I* and *siz1Δ*. Tetrads were dissected on YPD, then incubated at 30°C for 4 days. The offspring of one representative tetrad was shown with genotypes labeled. (**B**) Similar tetrad analysis for *rpc128-A704T* and *siz1Δ*. (**C**) Similar tetrad analysis for *rpc160-M809I* and *siz2Δ*. (**D**) 2 μg of RNA extracted from the indicated strains was run on a 2.8% agarose gel containing ethidium bromide, then visualized with UV. (**E**) RNA from (**C**) was reverse transcribed into cDNA, followed by real-time PCR analysis. GAPDH transcripts were used as loading control. Data are mean ± standard deviation calculated from six data points (two biological replicates and three technical replicates), presented as relative amount compared to wild type. The intron-containing pre-mature tRNA (pre-tL(CAA)A) is short-lived, so its abundance reflects the Pol III transcriptional activity.

DOI: https://doi.org/10.7554/eLife.35447.006

The following source data and figure supplements are available for figure 2:

**Source data 1.** Raw Ct values for *Figure 2E*.
DOI: https://doi.org/10.7554/eLife.35447.012

**Figure supplement 1.** Position of the mutated residues in Pol III structure.
DOI: https://doi.org/10.7554/eLife.35447.007

**Figure supplement 1—source data 1.** Source data for *Figure 2—figure supplement 1*
DOI: https://doi.org/10.7554/eLife.35447.008

**Figure supplement 2.** *siz1Δ* rescued a wide spectrum of Pol III mutations.
DOI: https://doi.org/10.7554/eLife.35447.009

**Figure supplement 3.** Growth phenotype of Pol III disease mutations in yeast.
DOI: https://doi.org/10.7554/eLife.35447.010

**Figure supplement 3—source data 1.** Source data for *Figure 2—figure supplement 3*
DOI: https://doi.org/10.7554/eLife.35447.011

## SUMO preferentially targets Pol III and acts independently of the known Pol III repressors

To gain further insights about the functions of SUMO in Pol III transcription, several specificity tests were performed. It is surprising that our screen only identified Pol III but not either of the other two polymerases, given the fact that the three polymerases are very similar to each other, with many related subunits and even shared subunits. We therefore first tested if it was due to a specific function of SUMO or simply because the screen was not saturated, by introducing similar mutations into the three RNA polymerases, such as an aspartic acid mutation to the glycine residue in the highly conserved 'trigger loop' domain in the largest subunits of the polymerases (*Fernández-Tornero et al., 2013*; *Hoffmann et al., 2015*; *Wang et al., 2006*) (*Figure 3A*). Interestingly, while the G to D mutation caused severe growth defect in all three cases, only the G1098D mutation in Rpc160 (Pol III) was rescued by *siz1Δ*, suggesting SUMO preferentially targets Pol III rather than Pol I or Pol II.

We next compared SUMO to known Pol III repressors, including Maf1 and the Mck1 and Kns1 kinases. Surprisingly, none of these proteins, when depleted by deleting the encoding genes, could rescue the *rpc160* mutant growth defect (*Figure 3B–D*). Furthermore, *siz1Δ* could rescue *rpc160* even in the absence of Maf1 (*Figure 3E*), and reverse the ability of *rpc160* to rescue *maf1Δ* on glycerol media (*Figure 3F*). Therefore, SUMO specifically targets Pol III for repression, and it does so through a mechanism that is independent of Maf1 or the Mck1 and Kns1 kinases.

## SUMO represses Pol III by modifying Rpc53

To understand the underlying molecular mechanism, the key is the relevant sumoylated protein(s), which is likely in the Pol III machinery itself. To identify this protein(s), we made a strain expressing Flag-tagged Rpc160-M809I and GFP-SUMO, as well as single-tagged strains as negative controls.

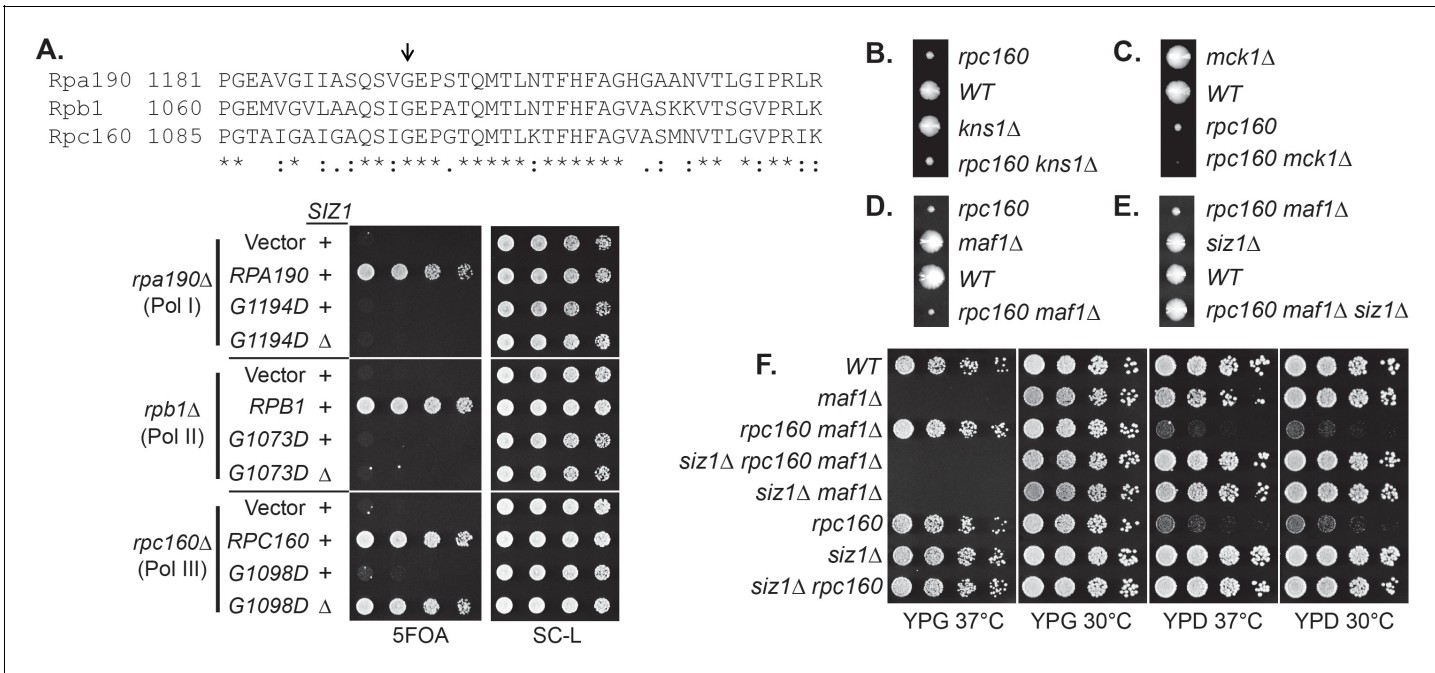

**Figure 3.** Specificity of the rescue effect. (**A**) The *rpa190Δ*, *rpb1Δ*, or *rpc160Δ* strain carries a *URA3* plasmid carrying wild-type *RPA190*, *RPB1*, and *RPC160* gene, respectively, in order to maintain viability. These strains were then transformed with *LEU2* plasmids carrying the indicated wild type or mutant alleles, and selected on synthetic media lacking leucine (SC-L). Transformants were spotted in fivefold serial dilutions onto a 5FOA plate to assess the growth phenotype of the mutant allele, as the original *URA3* plasmids were shuffled out of the cell in the presence of 5FOA. (**B–D**) Tetrad analysis between *rpc160-M809I* (shown as *rpc160*) and *kns1Δ*, *mck1Δ*, and *maf1Δ*. (**E**) Tetrad analysis between *rpc160-M809I siz1Δ* and *maf1Δ*. (**F**) The indicated strains from the cross in (**E**) were plated in fivefold dilutions onto YPD (glucose) or YPG (glycerol) plates and incubated at 30°C or 37°C as indicated.

DOI: https://doi.org/10.7554/eLife.35447.013

Mutant *rpc160-M809I* was used, because sumoylation has stronger effects on mutant Pol III than the wild type (*Figure 2D, E*). Pol III was first immunoprecipitated (IP) by anti-Flag beads (*Figure 4A*). Clear sumoylation signals were detected associated with Pol III. These Pol III-associated sumoylated proteins were released and subsequently purified by IP with GFP-trap beads, then analyzed by mass-spectrometry (*Figure 4B*). Four Pol III components (Rpc160, Rpc82, Rpc53, and Rpc37) were identified. Rpc160 (~160 kDa) is likely to be a contaminant, because the detected sumoylated species ran no slower than the 150 kDa marker band (*Figure 4A*). Rpc82 sumoylation was reported previously to occur on K406 (*Panse et al., 2004*). However, *rpc82-K406R* did not rescue *rpc160-M809I* (data not shown), suggesting Rpc82 is not the relevant sumoylated protein.

Rpc53 and Rpc37 form a subcomplex in the Pol III holoenzyme (*Hoffmann et al., 2015*; *Kassavetis et al., 2010*; *Landrieux et al., 2006*). Identifying both of them suggests that they are either tightly associated with a sumoylated protein or are sumoylated themselves. Indeed, Rpc53 was extensively sumoylated in vivo, and this was largely dependent on Siz1 but not Siz2 (*Figure 4C*), correlating with the fact that *siz2Δ* did not rescue the *rpc160* mutant growth defect (*Figure 2C*). Rpc53 was sumoylated more extensively in *rpc160* mutant cells (*Figure 4D*), suggesting that it may serve as a better SUMO substrate when Pol III is defective. The major sumoylation sites were mapped to K51, K115, and K236, by showing that mutating all three of them to arginines (*K51, 115, 236R*, or *rpc53-3KR*) abolished the majority of sumoylation, and no modification was detected when the N-terminal 274 amino acids of Rpc53 were deleted (Δ2–275) (*Figure 4E*). Importantly, *rpc53-3KR* rescued the *rpc128-A704T* growth defect (*Figure 4F*), whereas SUMO fusion to the N-terminus of Rpc53 (*Su-rpc53-3KR*), which mimics constitutive sumoylation, abolished the rescue effect of Rpc53-3KR. The SUMO-Rpc53-3KR fusion protein was expressed and functional, as it fully complemented the growth defect of *rpc53* null (*Figure 4G*). The rescue by *rpc53-3KR* was partial, suggesting additional modification sites in Rpc53 or other relevant SUMO substrates exist. Nevertheless, these results confirmed a direct relationship between SUMO and Pol III, and suggest that Rpc53 sumoylation can repress a defective Pol III machinery. Interestingly, the human homolog of Rpc53, RPC4, was found to be extensively sumoylated in mammalian cells by multiple proteomic studies (*Figure 4—figure supplement 1*), suggesting that this regulatory mechanism might be conserved from yeast to human.

## Pol III is repressed by ubiquitylation and the Cdc48 segregase

Sumoylation itself is not sufficient to inhibit Pol III and the effect of SUMO seems to be indirect, based on the facts that constitutive sumoylation of Rpc53 did not lead to any growth defect (*Figure 4G*), and that *rpc160-M809I* could also be rescued by a SUMO protease mutant, *ulp2-101* (*Figure 5—figure supplement 1A*), which did not abolish Rpc53 sumoylation (*Figure 5—figure supplement 1B*). Therefore, it is likely that Rpc53 sumoylation triggers a downstream event, such as STUbL-mediated ubiquitylation, which in turn represses Pol III. Indeed, the full repression of Pol III also requires ubiquitylation, as deletion of either one of the STUbL subunits (Slx5 and Slx8), or the Ubc4 ubiquitin E2 enzyme could all rescue the growth defect caused by *rpc160* mutations (*Figure 5A–C*). Furthermore, we could detect a physical interaction between Slx5 and Rpc53 in a yeast two-hybrid assay, which required Rpc53 sumoylation, as the 3KR mutation or the N-terminal deletion of Rpc53 abolished this interaction (*Figure 5D*). Consistently, the SUMO-interacting motifs (SIMs) in Slx5 are required for it to repress Pol III, as expression of the *slx5-sim* mutant, unlike wild-type *HA-SLX5*, did not reverse the rescue of *rpc160* mutations in the *slx5Δ* background (*Figure 5E*). These results are consistent with the STUbL activity of Slx5-Slx8 complex being important, and suggest that ubiquitylation acts downstream of sumoylation in Pol III repression.

Sumoylated and ubiquitylated proteins can both be targeted by Cdc48, leading us to test whether it is required in this case. As expected, *rpc160-M809I* was similarly rescued by the *cdc48-3* mutation (*Figure 5F*). However, the SIMs in Cdc48 (*Figure 5—figure supplement 2A*) or its cofactor Ufd1 (*Figure 5—figure supplement 2B*) were not required for its repressive effect, suggesting that Cdc48 activation does not occur through direct recognition of sumoylated Pol III complexes, but more likely through recognition of a ubiquitylated protein instead. The *ufd1-1* mutant has defects in the ubiquitin fusion degradation pathway (*Johnson et al., 1995*), but did not rescue *rpc160-M809I* (*Figure 5—figure supplement 2C*), suggesting Ufd1 is not the cofactor used by Cdc48 in this case.

To further explore the potential role of the Cdc48 segregase in Pol III complex disassembly, we performed tandem mass tag (TMT) mass spectrometry to determine the composition of Pol III

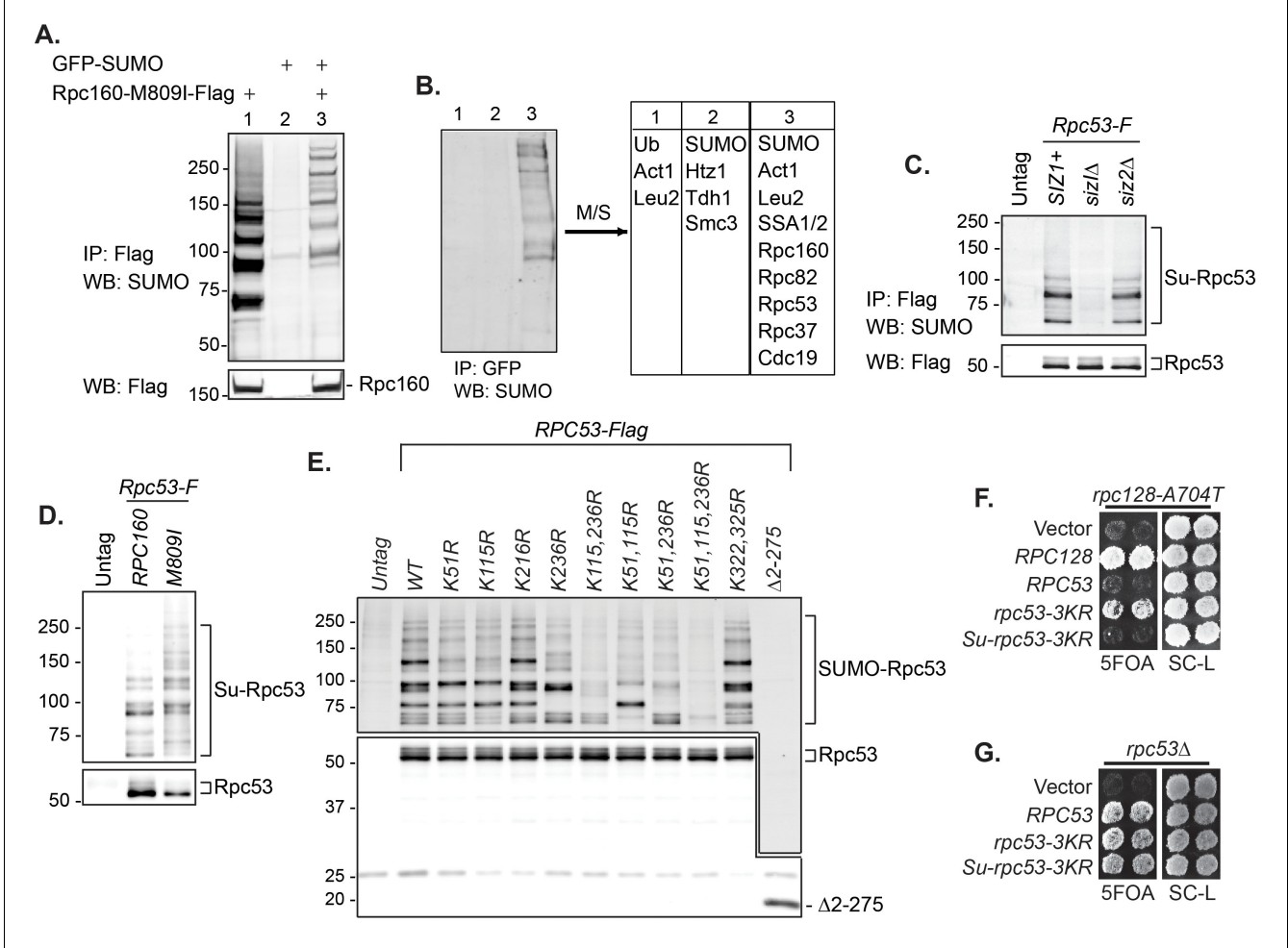

**Figure 4.** SUMO represses Pol III by modifying Rpc53. (**A**) Total protein extracted from the indicated strains was subjected to anti-Flag IP to purify Flag-tagged Rpc160 Pol III complexes and associated proteins. Precipitated proteins were eluted with Flag peptide, followed by SDS-PAGE and immunoblot analysis with an anti-Flag or anti-SUMO antibody. (**B**) The eluant from (**A**) was subjected to anti-GFP IP using GFP-Trap beads to isolate the sumoylated species from Pol III. The beads were subsequently washed with PBS containing 8M urea and 1% SDS to remove Rpc160-associated unmodified proteins, then incubated with 2× Laemmli's buffer at 100°C to elute sumoylated proteins. The success of the IP was confirmed by anti-SUMO immunoblot. The purified materials were subjected to tryptic digestion and analyzed by mass-spectrometry. (**C**) Flag-tagged Rpc53 was IP-ed from the indicated strains using anti-Flag beads, and detected by an anti-Flag antibody (bottom). Sumoylated Rpc53 (Su-Rpc53) was detected by anti-SUMO antibody (top). An untagged *RPC53* strain was used as a negative control. (**D**) Similar experiment as in (**C**) showing Rpc53 sumoylation in wild type *RPC160* cells versus *rpc160-M809I* mutant cells. (**E**) Mapping Rpc53 sumoylation sites by mutagenesis analysis. *CEN* plasmids carrying wild type or mutant Flag-tagged *RPC53* were co-transformed with a *2μ SMT3* plasmid into a wild-type yeast strain. Rpc53-Flag proteins were purified with anti-Flag IP, followed by SDS-PAGE and immunoblot analysis with anti-Flag (bottom) or anti-SUMO antibody (top). (**F**) An *rpc128-A704T* strain carrying a *URA3 RPC128* plasmid was transformed with *LEU2*-based *RPC128* or *RPC53* plasmids, then grown on 5FOA medium, which forces the cells to lose the *URA3 RPC128* plasmid. *rpc53-3KR* (K51,115,236R) rescued the growth of *rpc128-A704T*, whereas N-terminal SUMO fusion (*Su-rpc53-3KR*) abolished the rescue effect. The rescue effect is dominant because all the cells in this experiment contain wild type *RPC53* in the genome. (**G**) Similar plasmid shuffle experiment as in *Figure 2A*. The *LEU2* plasmids carrying the indicated *RPC53* alleles were transformed into an *rpc53Δ* strain containing a *URA3 RPC53* plasmid. The transformants were then plated onto a 5FOA plate to lose the *URA3 RPC53* plasmid, and the results showed that the N-terminally SUMO-fused Rpc53 protein (*Su-rpc53-3KR*) fully supports cell viability.

DOI: https://doi.org/10.7554/eLife.35447.014

The following source data and figure supplements are available for figure 4:

**Figure supplement 1.** Summary of sumoylated polymerase subunits.
DOI: https://doi.org/10.7554/eLife.35447.015
**Figure supplement 1—source data 1.** Source data for *Figure 4—figure supplement 1*
DOI: https://doi.org/10.7554/eLife.35447.016

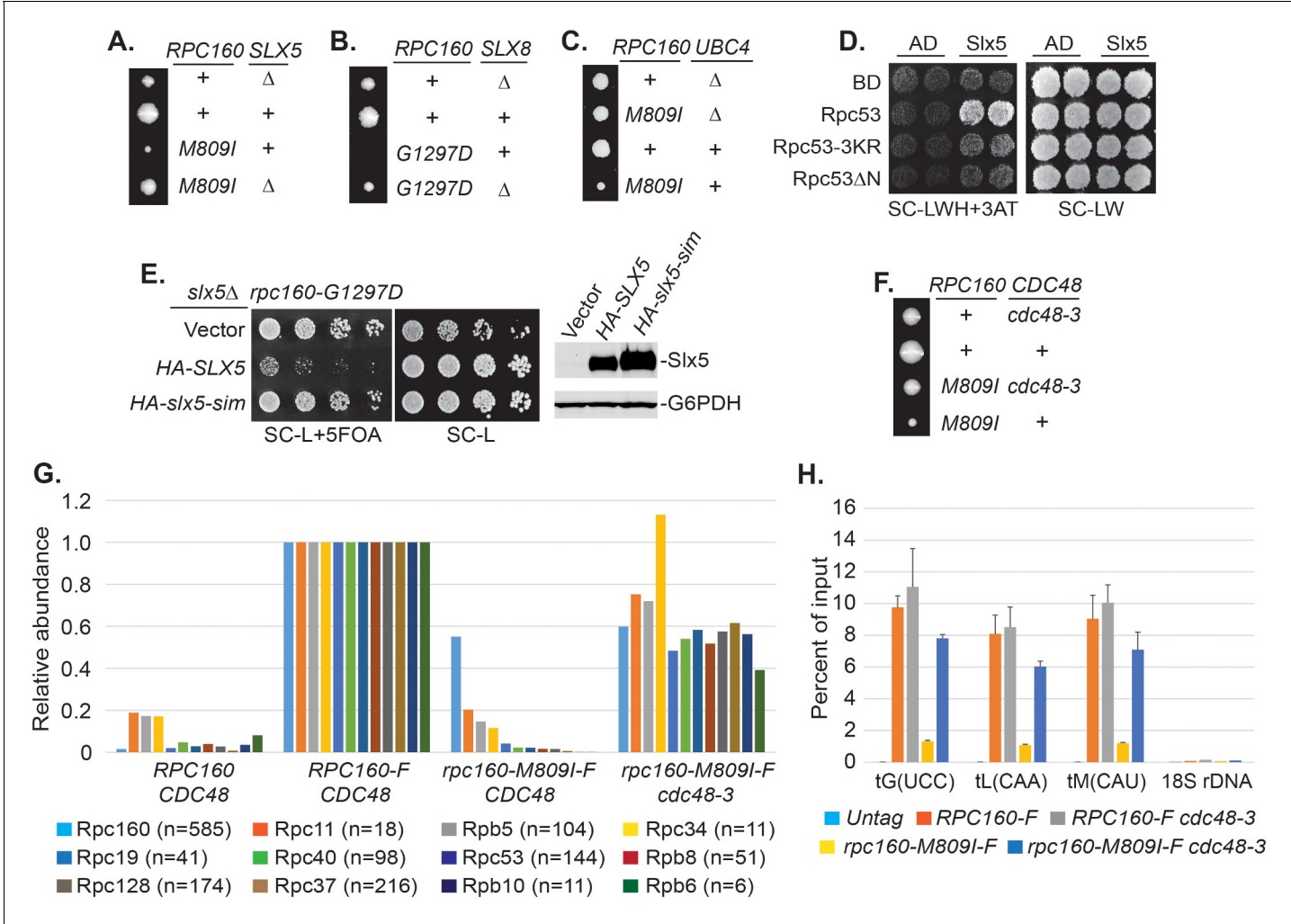

**Figure 5.** Pol III is repressed by ubiquitylation and p97/Cdc48. (**A–C**) The indicated *rpc160* mutant strains were crossed with *slx5Δ*, *slx8Δ*, or *ubc4Δ* strain, respectively, followed by tetrad analysis. The cross between *slx8Δ* and *rpc160-G1297D* was shown, because *slx8Δ* caused obvious growth defect by itself, so the rescue effect was more obvious on *rpc160-G1297D*, which is a sicker mutant than *rpc160-M809I*. (**D**) Yeast two-hybrid interactions between Slx5 and Rpc53. *SLX5* and *RPC53* were cloned into a 2μ *LEU2* Gal4 activation-domain (AD) vector and a 2μ *TRP1* DNA-binding domain (BD) vector, respectively, and co-transformed into yeast strain PJ69-4A. Transformants were selected on synthetic media lacking leucine and tryptophan (SC-LW), then patched and replica plated to selective media lacking histidine to test for interactions. The histidine-lacking media was supplement with 3-aminotriazole (SC-LWH + 3AT) for a more stringent phenotype. (**E**) *LEU2* plasmids carrying HA-tagged wild type or SIM-defective *SLX5* (*HA-slx5-sim*) were transformed into an *rpc160-G1297D slx5Δ* strain containing a *URA3 RPC160* plasmid. Transformants were selected on SC-L then spotted onto an SC-L + 5 FOA plate to lose wild-type *RPC160*. *HA-SLX5* complemented *slx5Δ* so the cells became sicker compared to the empty vector control transformants, while *HA-slx5-sim* did not complement, indicating that the SIMs are essential for the function of *SLX5* in this assay. The lost Slx5 function by the SIM mutations was not caused by insufficient proteins, since there were comparable levels of Slx5 proteins, as determined by an anti-HA immunoblot on total cell lysates (right, top panel). G6PDH served as a loading control (right, bottom panel). (**F**) Tetrad analysis between *rpc160-M809I* and *cdc48-3*. (**G**) Determination of Rpc160 association with other Pol III subunits. Pol III complexes containing Flag-tagged wild type or mutant Rpc160 in wild-type *CDC48* or *cdc48-3* cells were isolated using anti-Flag agarose beads, followed by TMS labeling and mass-spec analysis to quantify the relative amounts of Rpc160-interacting proteins. Signals for 12 of the 17 Pol III subunits, including Rpc160, were detected, and normalized to the signals from *RPC160-Flag* cells. n = Number of times when a unique peptide for the indicated protein is measured. An untagged *RPC160* strain was used as a negative control. (**H**) ChIP analysis of Rpc160. Flag-tagged *RPC160* or *rpc160-M809I* was expressed from a plasmid in wild-type *CDC48* or *cdc48-3* cells, as indicated. The Flag-tagged proteins were purified using anti-Flag agarose beads. An untagged strain was used as negative control. Three tRNA gene loci, as well as 18S rDNA (negative control), were examined. Chromatin association was determined by real-time PCR of the indicated genomic loci, using the percent of input method. Data are mean ± standard deviation calculated from six data points (two biological replicates and three technical replicates).

DOI: https://doi.org/10.7554/eLife.35447.017

The following source data and figure supplements are available for figure 5:

*Figure 5 continued on next page*

*Figure 5 continued*

**Source data 1.** Raw Ct values for *Figure 5G*.
DOI: https://doi.org/10.7554/eLife.35447.020
**Source data 2.** Raw Ct values for *Figure 5H*.
DOI: https://doi.org/10.7554/eLife.35447.021
**Figure supplement 1.** *ulp2-101* rescued *rpc160-M809I* without abolishing Rpc53 sumoylation.
DOI: https://doi.org/10.7554/eLife.35447.018
**Figure supplement 2.** Genetic relationship between *rpc160*, *cdc48*, and *ufd1*.
DOI: https://doi.org/10.7554/eLife.35447.019

complexes containing wild type or mutant Rpc160 (*Figure 5G*). Flag-tagged wild type or M809I mutant *RPC160* was ectopically expressed from a plasmid in wild-type *CDC48* or *cdc48-3* cells; all four strains in this experiment contain wild-type untagged Rpc160. The tagged proteins were immunoprecipitated by anti-Flag agarose beads under native conditions to preserve protein-protein interactions, followed by TMS labeling and MS analysis. Flag-tagged wild-type Rpc160 was found to interact with 11 of the 17 characterized Pol III subunits, and all of these were much more weakly associated with the Rpc160-M809I-Flag protein than with wild-type Rpc160-Flag. The reduced associations were largely restored in *cdc48-3* cells, which are deficient for Cdc48 segregase activity. These results suggest that Cdc48 may disassemble Pol III complexes that are transcriptionally defective. Consistently, chromatin immunoprecipitation (ChIP) experiments showed that compared with wild-type Rpc160, tRNA genes were significantly less occupied by mutant Rpc160-M809I, but the levels were restored in *cdc48-3* cells (*Figure 5H*), suggesting the defective Pol III complexes are removed from their target genes by Cdc48. However, there was no obvious increase of wild-type Rpc160 at tRNA genes in *cdc48-3* cells compared with wild-type *CDC48* cells, suggesting that only defective Pol III complexes are recognized and removed by Cdc48.

## Pol III repression is partially mediated by ubiquitylation of Rpc160

It is conceivable that STUbL-mediated ubiquitylation represses Pol III by modifying components of the transcription machinery, including subunits of Pol III itself. We noticed that the mutant Rpc160-M809I proteins are less stable than wild-type Rpc160, as determined by a cycloheximide-chase experiment (*Figure 6A*), suggesting the mutant Rpc160 proteins are degraded and therefore Rpc160 is likely to be ubiquitylated. Rpc160 can be ubiquitylated at K1240, K1242, K1249, K1273, and K1432, as determined by a previous proteomic study (*Swaney et al., 2013*). When the three clustered lysines were mutated to arginines (K1240, 1242, 1249R or 3KR), Rpc160-M809I proteins became more stable (*Figure 6A*) and the phenotypes of *rpc160-M809I*, including slow growth (*Figure 6B*) and reduced tRNA levels (*Figure 6C*), were partially rescued. The rescue effect was more obvious on the *rpc160-G1297D* mutant (*Figure 6D*). Interestingly, the 3KR mutation, when introduced into wild-type Rpc160 proteins, could rescue the defect caused by mutations in a different Pol III component, such as *rpc128-A704T* (*Figure 6E*). In addition, the Rpc160-M809I proteins were similarly stabilized in *siz1Δ*, *rpc53-3KR*, *ubc4Δ*, *slx8Δ* and *cdc48-3* cells (*Figure 6F*), as well as in the presence of a proteasome inhibitor, MG132 (*Figure 6G*). These results suggest that Rpc53 sumoylation leads to Rpc160 ubiquitylation by the Slx5-Slx8 STUbL, which subsequently triggers Pol III disassembly by the Cdc48 segregase, and eventually results in Rpc160 degradation by the proteasome. The rescue by 3KR is partial, suggesting other ubiquitylation sites in Rpc160 and/or additional ubiquitylated proteins exist that play a role.

## Sumoylation and ubiquitylation of Pol III occur on the chromatin

Chromatin association of SUMO at tRNA genes was previously reported in yeast (*Chymkowitch et al., 2017*) and mammalian cells (*Neyret-Kahn et al., 2013*). It is thus possible that sumoylation and ubiquitylation of Pol III subunits both occur on the chromatin. Indeed, we could detect an enrichment of Siz1 at tRNA genes (*Figure 7A*). Interestingly, more Siz1 protein was associated with tRNA genes in the mutant *rpc160-M809I* cells than in wild-type cells (*Figure 7A*, grey bars versus red bars), suggesting that the recruitment/association of Siz1 with tRNA genes might be triggered/stabilized by defective Pol III complexes. To further test this idea, we induced expression of mutant *rpc160-M809I* from a plasmid in wild-type *RPC160* cells, and asked what would happen

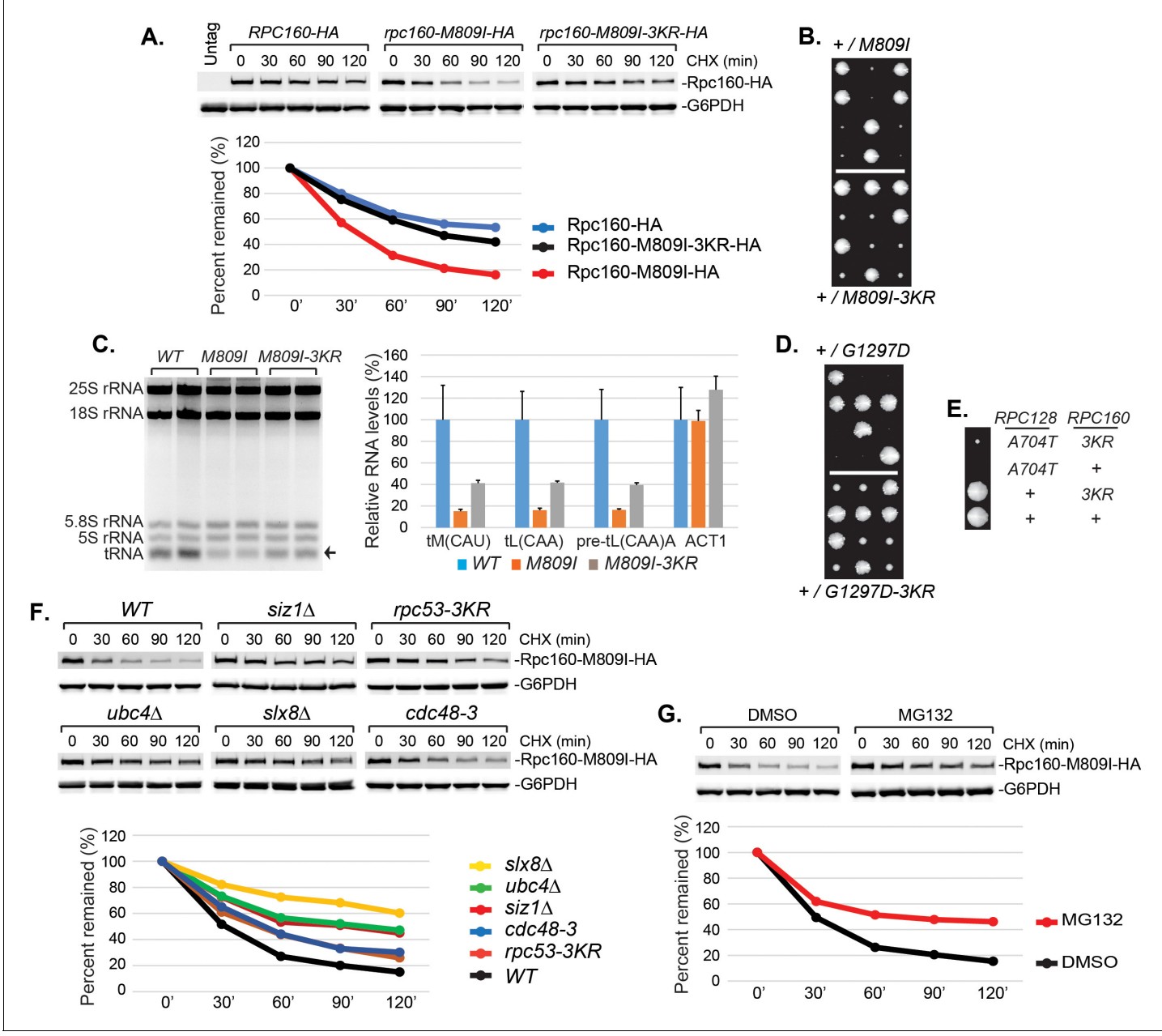

**Figure 6.** Pol III repression by ubiquitylation is partially mediated through Rpc160. (**A**) *CEN URA3* plasmids expressing HA-tagged wild type or mutant Rpc160, as indicated, were transformed into a wild-type strain, and their stabilities were assayed during a cycloheximide (CHX) chase time course. Rpc160 was detected by an anti-HA antibody, and G6PDH was used as a loading control. Quantification of the bands was shown below the immunoblot. (**B**) Tetrad analysis of the diploid strains, *RPC160+/rpc160-M809I* (top) and *RPC160+/rpc160-M809I-3KR* (bottom). Tetrads from these two diploids were dissected and plated on the same YPD plate at the same time, in order to compare the growth of *rpc160-M809I* and *rpc160-M809I-3KR* cells. The growth of three dissected tetrads were shown. The large colonies are wild-type *RPC160* cells, and the small colonies are *rpc160-M809I* (top) or *rpc160-M809I-3KR* (bottom) cells. The *rpc160-M809I-3KR* cells grew slightly faster than the *rpc160-M809I* cells. (**C**) Left: 2 µg of RNA extracted from the indicated strains was run on a 2.8% agarose gel containing ethidium bromide, then visualized with UV. Two colonies were picked for each strain. Right: RNA from left was reverse transcribed into cDNA, followed by real-time PCR analysis, as described in *Figure 2E*. GAPDH transcripts were used as loading control. (**D**) Similar tetrad analysis as in (**B**) of the diploid strains, *RPC160+/rpc160-G1297D* (top) and *RPC160+/rpc160-G1297D-3KR* (bottom). Large colonies are wild-type *RPC160* cells, while the missing colonies (top) are *rpc160-G1297D* cells, and the small colonies (bottom) are *rpc160-G1297D-3KR* cells. (**E**) An *rpc128-A704T* strain was crossed with an *rpc160-3KR* strain, followed by tetrad analysis. (**F**) A *CEN URA3 rpc160-M809I-HA* plasmid was transformed into the indicated strains, and the stabilities of the Rpc160-M809I-HA proteins were determined by CHX chase time course, as described in (**A**). (**G**) A *CEN URA3 rpc160-M809I-HA* plasmid was transformed into a wild-type strain, and protein stabilities were determined by CHX chase experiment in the presence of DMSO or MG132.

*Figure 6 continued on next page*

*Figure 6 continued*

DOI: https://doi.org/10.7554/eLife.35447.022

The following source data is available for figure 6:

**Source data 1.** Raw Ct values for *Figure 6C*.

DOI: https://doi.org/10.7554/eLife.35447.023

when the ratio of defective versus normal Pol III machinery increases in the cell. As shown in *Figure 7B*, left panel, more Siz1 was detected at tRNA genes upon induction of mutant *rpc160-M809I* expression in galactose media, compared to when additional wild-type *RPC160* expression was induced. Such a response was not observed when the expression of *rpc160-M809I* was repressed in glucose media (*Figure 7B*, right panel). These results are consistent with our hypothesis that the association of Siz1 with tRNA genes might be a response to disrupted Pol III transcription.

Besides Siz1, we were also able to detect an enrichment of Slx5 and Cdc48-3 at tRNA genes, especially in the mutant *rpc160-M809I* cells, similar to Siz1 (*Figure 7C, D*). Chromatin association of wild-type Cdc48 could not be detected (data not shown), possibly because it continuously disassembles the ubiquitylated Pol III complexes, releasing them as well as itself, from the chromatin. Surprisingly, Slx5 did not require its SIMs to associate with tRNA genes (*Figure 7C*), indicating the role of SUMO is not to recruit the STUbL to Pol III. Interestingly, however, the DNA-binding domain (SAP domain) of Siz1 (*Parker et al., 2008*; *Reindle et al., 2006*) was required for its tRNA gene association (*Figure 7A*). Furthermore, the SAP domain was also required for Siz1 to sumoylate Rpc53 (*Figure 7E*) and to inhibit the growth of the *rpc160* mutant cells (*Figure 7F*). Besides Siz1, the STUbL subunit Slx8 also contains a DNA-binding activity, which was mapped to the N-terminal 163 amino acids (*Yang et al., 2006*). Similarly, Slx8 requires this DNA-binding domain to inhibit the growth of the *rpc160* mutant cells (*Figure 7G*). However, the DNA-binding domain is not required for Slx8 to associate with chromatin (data not shown), as previously reported (*Yang et al., 2006*). Therefore, the targeting of Pol III by SUMO and ubiquitin is likely to occur on the chromatin, and require a physical interaction between DNA and the modifying enzymes.

## Discussion

Sumoylation is a very common posttranslational modification, with thousands of identified sumoylation sites in mammalian cells. Deletion of the *UBC9* SUMO-conjugating enzyme gene is an embryonic lethal event in mice underscoring the importance of sumoylation (*Nacerddine et al., 2005*). However, very few of the known sumoylation sites have been shown to have a functional consequence, because mutation of single sumoylation sites or even combinations in a protein usually results in no obvious phenotype. A possible explanation is provided by the 'SUMO spray' model, which proposes that a locally concentrated SUMO E3 ligase sumoylates multiple proteins nearby, allowing SUMO to serve as a glue for protein complex assembly (*Psakhye and Jentsch, 2012*). In such cases, sumoylation on multiple proteins would need to be abolished simultaneously to reveal a phenotype, making genetic analysis of SUMO function more difficult. To address this challenge, we devised, a phenotype-based genetic screen that selects for point mutations in yeast whose growth is rescued only when sumoylation is compromised, allowing identification of proteins where sumoylation has a functional consequence that can then be studied.

### SUMO-ubiquitin-Cdc48 is a new regulatory pathway for Pol III

We used a yeast genetic approach to uncover a functional relationship between Pol III and SUMO, demonstrating that genetics is a powerful tool to study sumoylation, complementary to the biochemical approach. Performing the same type of screen under different conditions is likely to yield more functional SUMO targets in the cell, and the same principle can be potentially extended to study other posttranslational forms. Our findings support a model where SUMO, ubiquitin, and Cdc48 act in a linear pathway to repress Pol III transcription (*Figure 8*). In this model, a defective/stalled chromatin-associated Pol III complex on a tRNA gene is first recognized by the chromatin-associated Siz1 E3 SUMO ligase and sumoylated on the Rpc53 subunit. Rpc53 sumoylation would then trigger ubiquitylation of the Rpc160 Pol III catalytic subunit, and possibly other proteins by the

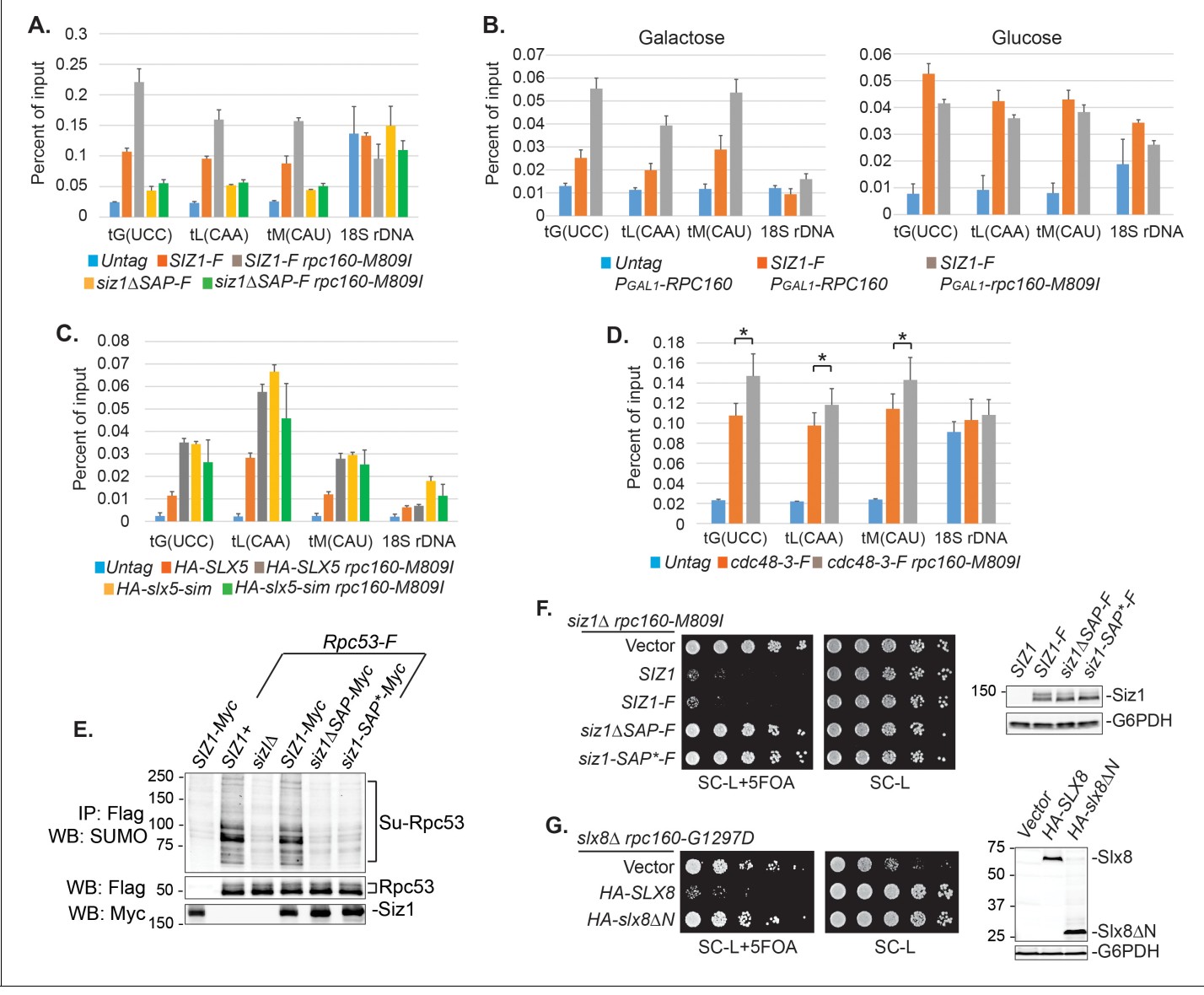

**Figure 7.** DNA is involved in Pol III repression. (**A**) Chromatin IP of Siz1-Flag. An untagged strain was used as negative control. Chromatin association was determined by real-time PCR of the indicated genomic loci, using the percent of input method. Data are mean ± standard deviation calculated from six data points (two biological replicates and three technical replicates). (**B**) Plasmids carrying *GAL1* promoter-driven *RPC160* or *rpc160-M809I* were transformed in untagged or Flag-tagged *SIZ1* cells as indicated. Cells were grown in galactose (left) or glucose (right) media to activate or inhibit transcription from the *GAL1* promoter, respectively, followed by ChIP analysis of Siz1-Flag. All strains also express wild-type *RPC160* from its endogenous promoter. (**C–D**) Similar ChIP analysis of HA-Slx5 and Cdc48-3-Flag as in (**A**). * p-value<0.05. (**E**) Flag-tagged Rpc53 proteins were purified from the indicated wild type or *siz1* mutant strains, using anti-Flag beads, followed by SDS-PAGE and immunoblotting using an anti-SUMO antibody. Rpc53 was detected by an anti-Flag antibody, and Siz1 was detected by an anti-Myc antibody. Either truncation (ΔSAP) or point mutation (SAP*) of the SAP domain resulted in loss of Rpc53 sumoylation. (**F**) Left: *LEU2* plasmids carrying wild type or mutant *SIZ1* alleles were transformed into an *rpc160-M809I siz1Δ* strain containing a *URA3 RPC160* plasmid. Transformants were selected on SC-L plate, then spotted in fivefold dilution onto a SC-L + 5 FOA plate. Wild type *SIZ1* complemented *siz1Δ* so the cells became sick on SC-L + 5 FOA plate, while the SAP mutants did not complement. Right: Comparable amounts of wild type and mutant Siz1 proteins were determined by anti-Flag immunoblotting on whole cell lysate, using G6PDH as loading control. (**G**) Similar plasmid shuffle experiment as in (**F**). *LEU2* plasmids carrying HA-tagged wild-type *SLX8* or *slx8ΔN* (Δ2–163) were transformed into an *rpc160-G1297D slx8Δ* strain containing a *URA3 RPC160* plasmid. Wild-type *SLX8* complemented *slx8Δ*, while *slx8ΔN* did not. Comparable amounts of Slx8 proteins were determined by an anti-HA immunoblot.

DOI: https://doi.org/10.7554/eLife.35447.024

The following source data is available for figure 7:

**Source data 1.** Raw Ct values for *Figure 7A–D*.
DOI: https://doi.org/10.7554/eLife.35447.025

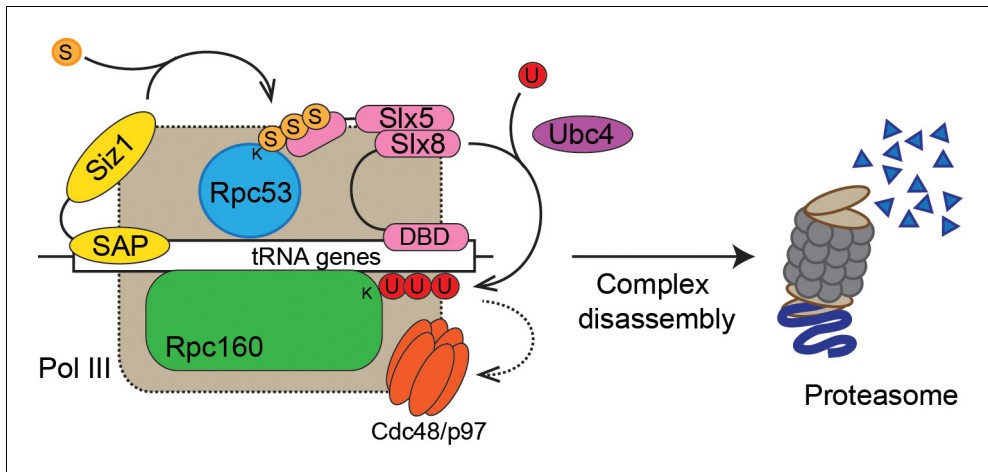

**Figure 8.** Model of Pol III regulation by SUMO, ubiquitin, and Cdc48. A stable interaction between chromatin and the SAP domain of Siz1 stimulates its activity to modify Rpc53 with SUMO (**S**). Rpc53 sumoylation triggers ubiquitin (**U**) modification of Rpc160 and potentially other proteins by the Slx5-Slx8 complex, which also required the interaction between chromatin and the DNA-binding domain (DBD) of Slx8. Ubiquitylation subsequently activates Cdc48 to disassemble the Pol III complex, facilitating degradation of Pol III subunits by the proteasome.
DOI: https://doi.org/10.7554/eLife.35447.026

chromatin-associated Slx5-Slx8 SUMO-targeted E3 ubiquitin ligase complex. Subsequently, ubiquity-lated Pol III complexes are recognized and disassembled by the Cdc48 AAA-ATPase segregase, leading to proteasomal degradation of the Rpc160 subunit, thus clearing obstructed tRNA genes to allow transcription to resume. However, as we were not able to directly detect Rpc160 ubiquityla-tion, most likely due to technical issues, alternative models of more complex interactions between SUMOylation, ubiquitylation, and segregation machineries in control of defective RNA pol III cannot be excluded. Nevertheless, we showed that this pathway is independent of the Mck1 and Kns1 kin-ases and Maf1, thus representing a new regulatory mechanism for Pol III. Interestingly, SUMO was recently shown to promote Pol III assembly and activity by modifying another subunit, Rpc82 (Chym-kowitch et al., 2017), suggesting that SUMO modification has complex regulatory effects on Pol III. By modifying different components of the Pol III machinery, SUMO may regulate Pol III at multiple transcription steps or in response to various signaling events.

### The SUMO-ubiquitin-Cdc48 pathway may serve as a quality control mechanism for Pol III

In the particular case reported here, the SUMO-ubiquitin-Cdc48 pathway seems to preferentially tar-get defective Pol III. For instance, *siz1Δ* only has an obvious effect when the function of Pol III is impaired (*Figure 2D, E*), and the *cdc48-3* mutant only increases the chromatin association of the defective but not wild-type Rpc160 protein (*Figure 5H*). Additionally, Siz1 can be recruited to tRNA genes in response to an elevated population of defective Pol III in the cell (*Figure 7B*), and maybe as a result, Rpc53 is more extensively sumoylated in mutant *rpc160* cells (*Figure 4D*). The question is, what Pol III defect is being recognized by the pathway? Since mutations in the Pol III initiation fac-tors Brf1, Tfc1, and Tfc6 can also be rescued by disrupting sumoylation, it is likely that a defect in transcription initiation is being recognized. This is supported by the result that *siz1Δ* rescued an initi-ation-defective mutant, *rpc31-236*. In Rpc160, the mutations T379I (close to the catalytic site), A880T (in bridge helix), and G1098T (in trigger loop) (*Hoffmann et al., 2015*) are likely to impair elongation, suggesting elongation defects might also be a feature recognized by the SUMO-ubiqui-tin-Cdc48 pathway. Consistently, two previously characterized elongation mutants, *rpc160-112* and *rpc160-270*, were both rescued by *siz1Δ*.

The next question is, how are initation- and elongation-defective Pol III proteins targeted by the pathway? Our data imply that it is the whole Pol III complex rather than individual subunits that are targeted for repression. For instance, sumoylated Rpc53 was co-purified with Rpc160, suggesting it

exists in the complex. In addition, when a mutation occurs in one subunit (e.g. Rpc128), SUMO and ubiquitin are conjugated to different subunits (Rpc53 and Rpc160, respectively), instead of Rpc128 itself, and the mutant Rpc128 proteins did not become unstable (data not shown). It is possible that SUMO and ubiquitin recognize overall conformational changes in the Pol III protein complex caused by a defective subunit, but we favor another possibility that involves chromatin DNA. It is conceivable that initiation or elongation defects will trap Pol III on the chromatin, forming a relatively stable protein/DNA complex. It would be a more efficient way to distinguish defective Pol III from normal Pol III molecules by utilizing the stable interaction between the E3 ligases and chromatin DNA, rather than by recognizing conformational changes. In fact, Siz1, Slx5, and Cdc48 are all associated with tRNA genes, and both Siz1 and Slx8 contain DNA-binding domains that are required for them to repress Pol III.

The requirement of the DNA-binding activities in Siz1 and Slx8 is somewhat surprising, since no functional consequences have been reported when they are disrupted. Specifically, the SAP-truncated version of Siz1 can still sumoylate PCNA and maintain the DNA-damage sensitivity of the *rad18* mutant, which can be reversed by complete deletion of *SIZ1* (*Parker et al., 2008*). Similarly, deleting the N-terminal DNA-binding domain of Slx8 does not affect its ability to associate with chromatin (Yang et al., 2006), or to complement the *slx8Δ* synthetic lethal phenotype with *sgs1Δ*. Therefore, the requirement for DNA-binding activity may indicate a role of SUMO and ubiquitin in a process other than the DNA damage response or genome stability maintenance.

Unlike the SAP domain in Siz1, the SIMs in Slx5 and the DNA-binding domain in Slx8 are not required for their chromatin association at tRNA genes, suggesting that their main function is not to recruit STUbL to Pol III. Instead, they may provide important docking sites to position the enzyme in the right orientation relative to the substrate in order for it to ubiquitylate a specific target subunit in Pol III. It is thus possible that the STUbL can travel with sumoylated Pol III without ubiquitylating the polymerase until Pol III is somehow trapped on the chromatin, which will allow STUbL to stably bind to DNA and activate its ligase activity. Therefore, Rpc53 sumoylation itself will not be sufficient to trigger ubiquitylation to inhibit Pol III. This is supported by the finding that SUMO-fused Rpc53 did not affect cellular growth (*Figure 4G*), even though the N-terminal SUMO fusion could functionally replace sumoylation on the natural modification sites (K51, K115, and K236) (*Figure 4F*). These results also suggest that sumoylation may not activate Cdc48 directly, but rather indirectly through ubiquitylation. Consistently, the SUMO-interacting activities of Cdc48 and its cofactor Ufd1 were not required for Pol III repression. Our data suggest that Cdc48 may be able to disassemble defective Pol III complexes (*Figure 5G*) and remove them from the chromatin (*Figure 5H*), although how Cdc48 is activated and what cofactors are required for Cdc48 in this case remain to be answered.

Taken together, we propose that SUMO, ubiquitin, and Cdc48 act in a sequential manner, and that together with the additional requirement of DNA-binding activities, they confer substrate specificity to restrict ubiquitylation, as well as subsequent complex disassembly and proteasomal degradation, only toward transcriptionally defective Pol III, while leaving normal Pol III unaffected. To further test our hypothesis, in vitro sumoylation, ubiquitylation, and Pol III transcription assays with purified proteins are required.

## The SUMO-ubiquitin-Cdc48 pathway is a potential target for Pol III-related human diseases

Given the fact that Pol III, sumoylation, ubiquitylation, and Cdc48 are all conserved from yeast to humans, this new Pol III regulatory mechanism is likely to be conserved as well. In fact, proteomic studies of sumoylation in mammalian cells have identified sumoylated proteins in Pol III, including four Pol III-specific subunits and two subunits shared by Pol I and/or Pol II (*Figure 4—figure supplement 1*) (*Hendriks et al., 2014*; *Lamoliatte et al., 2014*; *Tammsalu et al., 2014*). By comparison, only three subunits for each of Pol I and Pol II were found to be sumoylated, suggesting Pol III is the major SUMO target among the three polymerases. This correlates with the specific genetic relationship between SUMO and Pol III observed in yeast, namely that *siz1Δ* only rescued mutations in Pol III but not those in Pol I or Pol II. Interestingly, RPC4, the human homologue of yeast Rpc53, seems to be the most prevalent sumoylated protein among all human RNA polymerase subunits, with a total number of 10 sumoylation sites combining all three datasets.

Mutations in Pol III cause neurodegenerative disorders in humans (*Bernard et al., 2011*; *Borck et al., 2015*; *Saitsu et al., 2011*; *Shimojima et al., 2014*; *Synofzik et al., 2013*; *Terao et al.,*

*2012*; *Tétreault et al., 2011*; *Thiffault et al., 2015*). We showed, interestingly, that the phenotypes caused by a subset of these Pol III disease mutations, when introduced into yeast cells, were rescued when sumoylation was disrupted (*Figure 2—figure supplement 2* and *Figure 2—figure supplement 3*). In addition, the fibroblasts derived from a patient carrying Pol III mutations exhibited reduced levels of tRNA (data not shown), as is the case in yeast. It is thus intriguing to speculate that sumoylation, ubiquitylation, and Cdc48 segregase can all be potential therapeutic targets for the neurodegenerative disease caused by Pol III mutations. The next step is to determine if this regulatory mechanism is conserved in humans, by testing it in cultured mammalian cells and mouse model systems. We have shown that the yeast growth assay is a convenient tool to determine which neurodegenerative disease mutations are likely to cause a phenotype and therefore, which should be chosen to create cell lines and mouse models. It is possible to differentiate iPSC lines into myelinating oligodendrocytes in vitro (*Kerman et al., 2015*), and the oligodendrocytes from Pol III patients are expected to display a myelination defect. Alternatively, a cell line can be made to carry a lethal mutation in one copy of a Pol III subunit gene while leaving the wild-type copy under control of an inducible promoter. Such a cell line can be adapted for high-throughput chemical screens for inhibitors against sumoylation, ubiquitylation, or Cdc48, which can not only be used as research tools, but also be developed into potential therapies for Pol III-related disorders or other human diseases involving SUMO, ubiquitin, or Cdc48, such as cancer (*Kessler et al., 2012*). A genetically modified mouse model will eventually be needed to recapitulate the disease and test the effect of the inhibitors.

# Materials and methods

## Key resources table

| Reagent type (species) or resource | Designation | Source or reference | Identifiers | Additional information |
|---|---|---|---|---|
| Antibody | anti-HA | Santa Cruz | SC-7392 | |
| Antibody | anti-HA beads | Sigma | A2095 | |
| Antibody | anti-Flag | Sigma | F3165 | |
| Antibody | anti-Flag beads | Sigma | A2220 | |
| Antibody | anti-Myc | This study | | |
| Antibody | anti-G6PDH | Sigma | A9521 | |
| Antibody | anti-Smt3 (SUMO) | Santa Cruz | SC-28649 | |
| Antibody | GFP-Trap beads | chromotek | gta-20 | |
| Antibody | Alexa Fluor 680 Goat anti-Mouse IgG | Fisher Scientific | A21058 | |
| Antibody | Alexa Fluor 680 Goat anti-Rabbit IgG | Fisher Scientific | A21109 | |
| Antibody | DyLight 800 Goat anti-Rabbit IgG | Fisher Scientific | SA535571 | |
| Antibody | DyLight 800 Goat anti-Mouse IgG | Fisher Scientific | SA535521 | |
| Peptide, recombinant protein | 2x HA peptide | This study | | |
| Peptide, recombinant protein | 2x Flag peptide | This study | | |
| Commercial assay or kit | cOmplete, Mini, EDTA-free Protease Inhibitor Cocktail | Sigma | 11836170001 | |

*Continued on next page*

*Continued*

| Reagent type (species) or resource | Designation | Source or reference | Identifiers | Additional information |
|---|---|---|---|---|
| Commercial assay or kit | Power SYBR Green PCR Master Mix | Fisher Scientific | 4367659 | |
| Commercial assay or kit | SuperScript III First-Strand Synthesis System | Invitrogen | 18080–051 | |
| Chemical compound, drug | N-Ethylmaleimide (NEM) | Sigma | E3876 | |
| Chemical compound, drug | 5-Fluoroorotic Acid (5-FOA) | Toronto Research Chemicals | F595000 | |
| Chemical compound, drug | MG132 | Fisher Scientific | 50-833-9 | |
| Chemical compound, drug | Cycloheximide | Sigma | C7698 | |
| Software, algorithm | Excel | Microsoft | | |

## Yeast strains, plasmids, media, and genetic methods

The *Saccharomyces cerevisiae* strains and plasmids used in this study are listed in *Supplementary file 1* and *Supplementary file 2*, respectively. All ZBY strains are in S288C background. All media used, including rich YPD medium (yeast extract-peptone-dextrose), sucrose medium (yeast extract-peptone-Suc), synthetic complete (SC) drop-out medium (for example, SC-U), SC-galactose medium and sporulation medium, were made as described previously (*Rose et al., 1990*). SC-L + 5 FOA plates were made as standard SC drop-out medium, but using 2 g of SC-UL drop-out mix, plus 50 mg of uracil, and 1 g of 5FOA per 1 liter of total volume. For a more stringent yeast two-hybrid interaction signal, 24 μmol of 3-aminotriazole (3AT) was spread onto a 10 cm SC-LWH plate. Standard genetic methods for mating, sporulation, transformation, and tetrad analysis were used throughout this study. In the tetrad analysis experiments, the mutant haploid *rpc160* or *rpc128* strains contain a *URA3* vector carrying wild type copy of gene, in order to maintain the strains. The *URA3* plasmids were lost from the diploid cells on 5FOA media after mating and before sporulation. The similar strategy was used in the plasmid shuffle experiments, for example, in *Figure 3A*, where the starting *rpc160Δ* strain contains a *URA3 RPC160* plasmid. Upon transformation with *LEU2* plasmids carrying *rpc160* mutant alleles, the transformants were plated onto 5FOA-containing media to lose the *URA3 RPC160* plasmid. Growth on 5FOA media therefore reflects the growth phenotype of the *rpc160* mutants present on the *LEU2* plasmids.

The mutations in *slx5-sim* are: 24VILI – VAAA, 93ITII – ATAA, 116YVDL – YAAA, and 155LTIV – ATAA. The *siz1ΔSAP* and *siz1-SAP\** mutations were made as previously described (*Parker et al., 2008*).

## Design of the reverse suppressor screen of *SMT3-Q56K*

The starting strain is ZOY261 (*ade2 ade3 ura3 leu2 trp1 can1Δ::FUR1::natMX4*) carrying two plasmids, pZW321 (*CEN URA3 ADE3 SMT3-Q56K*) and pAK12-1 (*CEN TRP1 ade3-pink*) (*Koren et al., 2003*), and grown on SC-Leu-Trp in order to keep the plasmids. Wild type yeast cells are white, while *ade2* mutant is red. *ade3* mutation suppresses *ade2*, so that *ade2 ade3* double mutant is white. Both *ade2* and *ade3* are recessive, so in the presence of pZW321, ZOY261 colonies are red. The starting strain (ZOY261 + pZW321 + pAK12-1), if grown on YPD, does not need pZW321 for viability, so the cells will lose the plasmid during cell proliferation, eventually forming colonies with red and white sectors. The screen looks for mutations that cause sickness or lethality, but can be suppressed/rescued by *SMT3-Q56K* on pZW321. These mutants will appear as uniformly red colonies, because they always need to keep pZW321 for viability. They will also be sensitive to 5FOA, because 5FOA counter-selects against *URA3* on pZW321.

To perform the screen, cells were mutagenized with 3% ethyl methanesulfonate (EMS), washed, and then spread onto YPD plates at 30°C to allow formation of single colonies. Uniformly red colonies were first picked and re-streaked on fresh YPD plates. The clones remaining uniformly red after

restreak were subsequently screened for those that are 5FOA-sensitive. The red 5FOA-sensitive colonies could also come from mutations that are synthetic lethal with the *ura3* or *ade3* alleles, as pZW321 also carries wild-type *URA3* and *ADE3*. To reduce the chance of isolating these undesired mutations, two modifications were made. First, an additional copy of wild-type *FUR1* gene was integrated at the *CAN1* gene locus of the starting strain, because *ura3* synthetic lethal mutations are most frequently found in *FUR1* (*Koren et al., 2003*). Second, the pAK12-1 plasmid carrying an *ade3-pink* allele was co-transformed with pZW321 into ZOY261. Unlike wild-type *ADE3*, the *ade3-pink* allele is partially functional, but confers a pink (instead of red) colony color phenotype in *ade2 ade3* background (*Koren et al., 2003*). Therefore, in the presence of pAK12-1, the *ade3* synthetic lethal mutants will not appear as uniformly red colonies, but sectored with red and pink instead. To finally confirm that a strain contains a *SMT3-Q56K*-rescuable mutation, the strain was transformed with the pZW311 plasmid (*CEN LEU2 SMT3-Q56K*), which should render the cells resistant to 5FOA after transformation.

To identify the mutated gene, a wild-type genomic DNA library (*Jones et al., 2008*) was transformed into the candidate strains, and screened for 5FOA-resistant transformants. The plasmids were then isolated and sequenced to identify the ends of the genomic DNA on the plasmids. The mutated genes were identified by subcloning or by complementation experiments. The genomic mutations were finally confirmed by PCR and sequencing. To summarize, ~80,000 colonies were initially screened, and 740 uniformly red ones were picked and restreaked. One hundred and five clones remained red after restreak, among which 77 were 5FOA-sensitive. Finally, 25 of these 77 clones were confirmed to have mutations that can be suppressed by *SMT3-Q56K* (*Table 1*).

## Preparation of RNA from yeast cells

RNA was prepared by the 'Heat/Freeze' method as previously described with modifications (*Schmitt et al., 1990*). Briefly, yeast cells were resuspended in AE buffer (50 mM NaOAc pH 5.2, 10 mM EDTA, 1% SDS), mixed with equal volume of phenol (pH 4.5), then incubated at 65°C for 4 min. The cell suspension was then frozen on dry ice/ethanol bath and thawed at 37°C. After centrifugation at top speed, the RNA containing upper layer was transferred to a new tube. RNA was extracted first with phenol/chloroform/isoamyl alcohol (25:24:1), then with chloroform/isoamyl alcohol (24:1), and finally precipitated with 100% ethanol containing 0.3M NaOAc (pH 5.2). The RNA pellet was washed once with 70% ethanol and once with 100% ethanol, then dissolved in DEPC-treated $H_2O$ at 50°C for 10 min.

## Reverse transcription and real-time PCR analysis

RNA was converted into DNA using the SuperScript III First Strand Synthesis System (Invitrogen, catalog # 18080–051) with modifications. First, a mixture of random hexamer and tRNA gene-specific primers (*Supplementary file 3*) was used for reverse transcription. Second, primers were hybridized to RNA by incubating the sample at 100°C for 5 min, followed by 65°C for 5 min, then held at 55°C. Third, the RT enzyme mix was pre-warmed to 55°C before adding to the RNA/primer mix. Fourth, reverse transcription was carried out at 55°C for 30 min, followed by 85°C for 5 min, and finally held at 4°C. The resulting DNA was diluted 10 times with $H_2O$, and 2 μl of the diluted DNA was used for real-time PCR. Real-time PCR was performed with SYBR Green master mix (Applied Biosystems, catalog # 4367659) on the Applied Biosystems 7900HT Fast Real-Time PCR System. Data were analyzed by the comparative $C_T$ method (*Schmittgen and Livak, 2008*), using the average Ct value of GAPDH as internal control. Primer sequences were listed in *Supplementary file 3*. Data were analyzed using Excel and presented as mean ± standard deviation calculated from six data points (two biological replicates and three technical replicates), presented as relative amount compared to wild type.

## Preparation of protein extracts and immunoprecipitation (IP)

For IPs, crude protein extracts were prepared by the glass bead beating method, as described above (*Wang and Prelich, 2009*). Briefly, cells were first resuspended in lysis buffer containing 50 mM Tris-Cl (pH 8.0), 10 mM $MgCl_2$, 1 mM EDTA, 150 mM NaCl, 1 mM phenylmethylsulfonyl fluoride (PMSF), 50 mM *N*-ethylmaleimide (NEM), 1% Triton X-100, and protease inhibitors (Roche, catalog # 11836170001), and glass beads. Cell lysis was subsequently performed on a Precellys 24 tissue homogenizer (setting 6500, 30 s for three times), followed by centrifugation to clear the lysate. IP

was performed with anti-HA agarose (Sigma, catalog #A2095) or anti-Flag-agarose (Sigma, #A2220) at 4°C for a typical length of 2 hr, and the bound proteins were eluted with 2x HA or 2x Flag peptide, respectively. HA- or Flag-tagged protein was detected with anti-HA (Santa Cruz, #SC-7392) or anti-Flag (Sigma, #F3165), respectively.

For the CHX chase experiments in *Figure 6* and the immunoblotting in *Figure 7F, G*, protein extracts were prepared by the post-alkaline extraction method, as previously described (*Zhang et al., 2011*). Briefly, 1–2 O.D. of cells were first washed with 1 ml of water, then resuspended in 200 µl 2M LiOAc and incubated on ice for 5 min. The cells were subsequently resuspended in 200 µl 0.4M NaOH and incubated on ice for another 5 min, before finally being resuspended in 40 µl Laemmli's buffer and incubated at 100°C for 5 min. Of protein sample, 10 µl were loaded on the gel.

## Assays of protein stability

Yeast cultures were grown overnight at 30°C to log phase. Cell concentrations were then adjusted to OD600 = 1. To start the chase, 50 mg/ml cycloheximide (Sigma, #C7698) was added to a final concentration of 0.5 mg/ml. 1.5 ml culture was collected immediately as time point 0 in an Eppendorf tube pre-loaded with 15 µl 10% sodium azide. Cells were then washed with 1 ml water and frozen on dry ice. The remaining cultures were incubated at 30°C, and 1.5 ml samples were collected every 30 min in the same way. Crude extracts were prepared by the post-alkaline extraction method as described above. An anti-G6PDH (Sigma, #A9521) antibody was used to detect G6PDH as loading control. Immunoblot analysis was performed using the Odyssey infrared imaging system (*LI-COR* Biosciences). Fluorescent dye-conjugated secondary antibodies used were listed in Key Resources Table.

## Two-step immunoprecipitation to isolate Pol III-associated sumoylated proteins

Flag-tagged Rpc160 was first purified by incubating with anti-Flag M2 affinity gel (Sigma, catalog # A2220) in lysis buffer, then eluted with 450 ng/µl 2x Flag peptide in GFP-IP buffer (50 mM Tris-Cl (pH 7.5), 1 mM EDTA, 500 mM NaCl, 1 mM PMSF, 50 mM NEM, and protease inhibitors). For the second step, the Flag-eluted protein sample was incubated with GFP-Trap agarose beads (Chromo-Tek, catalog # gta-10), followed by two washes with GFP-IP buffer, one quick wash with PBS containing 8 M urea and 1% SDS, and one with PBS containing 1% SDS. GFP-tagged proteins were eventually eluted with 2x Laemmli's buffer (without dye) at 100°C for 5 min, and analyzed by mass-spectrometry. Immunoprecipitated samples were analyzed by SDS-PAGE, followed by immunoblotting with an anti-SUMO antibody (Santa Cruz, Smt3 (y-84), catalog # sc-28649) or an anti-Flag M2 antibody (Sigma). The *SLX5* gene was deleted from the strains in *Figure 3A*, in order to increase general sumoylation signal (*Wang et al., 2006*; *Wang and Prelich, 2009*).

## Mass spectrometry analysis

Samples were first denatured in 8 M urea and then reduced and alkylated with 10 mM Tris (2-carboxyethyl) phosphine hydrochloride (Roche Applied Science) and 55 mM iodoacetamide (Sigma-Aldrich), respectively. Samples were then digested over-night with trypsin (Promega) according to the manufacturer's specifications. The protein digests were pressure-loaded onto 250 micron i.d. fused silica capillary (Polymicro Technologies) columns with a Kasil frit packed with 3 cm of 5 micron C18 resin (Phenomenex). After desalting, each loading column was connected to a 100 micron i.d. fused silica capillary (Polymicro Technologies) analytical column with a five micron pulled-tip, packed with 12 cm of 5 micron C18 resin (Phenomenex).

Each split column was placed in line with an 1100 quaternary HPLC pump (Agilent Technologies) and the eluted peptides were electrosprayed directly into an Orbitrap Elite mass spectrometer (Thermo Scientific). The buffer solutions used were 5% acetonitrile/0.1% formic acid (buffer A) and 80% acetonitrile/0.1% formic acid (buffer B). The 120 min elution gradient had the following profile: 10% buffer B beginning at 10 min to 45% buffer B at 90 min, and then 100% buffer B at 100 min continuing to 110 min. A cycle consisted of one full scan mass spectrum (300–1600 m/z) in the Orbitrap at 120,000 resolution followed by 15 data-dependent collision induced dissociation (CID) MS/MS spectra in the ion trap. Charge state screening was enabled and unassigned charge states and

charge state one were rejected. Dynamic exclusion was enabled with a repeat count of 1, a repeat duration of 30 s, an exclusion list size of 500 and an exclusion duration of 120 s. Dynamic exclusion early expiration was enabled with an expiration count of 3 and an expiration signal-to-noise ratio of 3. Application of mass spectrometer scan functions and HPLC solvent gradients were controlled by the Xcalibur data system (Thermo Scientific).

MS/MS spectra were extracted using RawXtract (version 1.9.9.2) (*McDonald et al., 2004*). MS/MS spectra were searched with the ProLuCID (version 1.3.5) algorithm (*Xu et al., 2015a*) against a Saccharomyces Genome Database (SGD) protein database downloaded on 01-05-2010 that had been concatenated to a decoy database in which the sequence for each entry in the original database was reversed (*Peng et al., 2003*). A total of 13,434 protein entries were searched. Precursor mass tolerance was 50 ppm and fragment mass tolerance was 600 ppm. For protein identifications, the ProLuCID search was performed using no enzyme specificity and static modification of cysteine due to carboxyamidomethylation (57.02146). ProLuCID search results were assembled and filtered using the DTASelect (version 2.1.3) algorithm (*Tabb et al., 2002*), requiring full enzyme specificity (cleavage C-terminal to Arg or Lys residue) and a minimum of one peptide per protein identification. The number of missed cleavages was not specified. The protein identification false positive rate was kept below 1% and all peptide-spectra matches had less than 10 ppm mass error. DTASelect assesses the validity of peptide-spectra matches using the cross-correlation score (XCorr) and normalized difference in cross-correlation scores (deltaCN). The search results are grouped by charge state and tryptic status and each sub-group is analyzed by discriminant analysis based on a non-parametric fit of the distribution of forward and reversed matches.

## Tandem Mass Tag (TMT) labeling and mass spectrometry analysis

Sample were precipitated using methanol- chloroform. Dried pellets were dissolved in 8 M urea, reduced with 5 mM Tris (2-carboxyethyl) phosphine hydrochloride (TCEP), and alkylated with 50 mM chloroacetamide. Proteins were then trypsin digested overnight at 37°C. The digested peptides were labeled with TMT10plex Isobaric Label Reagent Set (Thermo catalog number 90309 (lot SK259407) and fractionated by basic reverse phase HPLC (Thermo 84868). The TMT labeled samples were analyzed on a Fusion Lumos mass spectrometer (Thermo). Samples were injected directly onto a 25 cm, 100 μm ID column packed with BEH 1.7 μm C18 resin (Waters). Samples were separated at a flow rate of 200 nL/min on a nLC 1200 (Thermo). Buffer A and B were 0.1% formic acid in water and 90% acetonitrile, respectively. A gradient of 1–20% B over 280 min, an increase to 40% B over 50 min, an increase to 100% B over another 20 min and held at 90% B for a final 10 min of washing was used for 360 min total run time. Columns were re-equilibrated with 20 μL of buffer A prior to the injection of sample. Peptides were eluted directly from the tip of the column and nanosprayed directly into the mass spectrometer by application of 2.8 kV voltage at the back of the column. The Lumos was operated in a data-dependent mode. Full MS1 scans were collected in the Orbitrap at 120 k resolution. The cycle time was set to 3 s, and within this 3 s the most abundant ions per scan were selected for CID MS/MS in the ion trap. MS3 analysis with multinotch isolation (SPS3) was utilized for detection of TMT reporter ions at 15 k resolution. Monoisotopic precursor selection was enabled and dynamic exclusion was used with exclusion duration of 10 s.

Protein and peptide identifications were done with Integrated Proteomics Pipeline – IP2 (Integrated Proteomics Applications). Tandem mass spectra were extracted from raw files using RawConverter (*He et al., 2015*) and searched with ProLuCID (*Xu et al., 2015b*) against *Saccharomyces cerevisiae* Uniprot database (UP000002311 released June 7, 2015). The search space included all fully-tryptic candidates. Carbamidomethylation on cysteine and TMT labels on N terminus and lysine were considered as static modifications. Data were searched with 50 ppm precursor ion tolerance and 600 ppm fragment ion tolerance. Identified proteins were filtered using DTASelect (*Tabb et al., 2002*) and utilizing a target-decoy database search strategy to control the protein false discovery rate to 1% (*Peng et al., 2003*). Quantification was done using Census (*Park et al., 2014*). Values were normalized to Rpc160.

## Chromatin immunoprecipitation (ChIP) and real-time PCR analysis

ChIPs were performed as previously described (*Keogh et al., 2003*). Yeast cells were grown over night at 30°C to log phase. Formaldehyde was added to a final concentration of 1% for 20 min at

room temperature, and the reaction was quenched by the addition of glycine to 0.3 M. Cells were washed twice with Tris-buffered saline, and lysed with glass beads in FA lysis buffer containing 50 mM Tris-Cl (pH 8.0), 10 mM $MgCl_2$, 1 mM EDTA, 150 mM NaCl, 1 mM phenylmethylsulfonyl fluoride (PMSF), 50 mM *N*-ethylmaleimide (NEM), 1% Triton X-100, 0.1% SDS, 0.1% sodium deoxycholate, and protease inhibitors (Roche, catalog # 11836170001). Chromatin was sheared by sonication until the average fragment size was between 200 and 500 bp. 100 µl and 700 µl of crude chromatin samples were used as input and to perform immunoprecipitation, respectively.

Immunoprecipitation was performed over night at 4°C with anti-HA or anti-Flag agarose. The bound materials were eluted with 2x HA or 2x Flag peptide, followed by RNase A and protease K treatment to de-crosslink. Bound DNA was then purified with the Qiagen PCR purification kit, and analyzed by real-time PCR, as described above. Data are mean ± standard deviation calculated from six data points (two biological replicates and three technical replicates), using the percent of input method. An untagged strain was used as negative control. Statistical analysis of differences between two groups was performed using a two-tailed, equal variance t-test (Excel).The sequences of the primers were listed in *Supplementary file 3*.

## Acknowledgements

We thank Gregory Prelich for the SUMO pathway mutant plasmids and strains and the wild type genomic DNA library, Martin Kupiec for the *ade3-pink* plasmid, Michel Werner for the *rpc160* and *rpc31* mutant plasmids, Vicki Lundblad for sharing yeast vectors and equipment, and Jill Meisenhelder for the 2x Flag peptide. We also thank James Moresco and Jolene Diedrich for technical support on TMT-M/S analysis, Benoit Coulombe and Genevieve Bernard for sharing unpublished data on Pol III-related diseases, as well as Ian Willis, Alessandro Vannini, Gregory Prelich, and members of the Hunter lab for constructive discussions. This study was supported by NIH grants #CA080100, CA082683 and CA014195 to TH, NCRR grant #5P41RR011823-17 and NIGMS grant #8P41GM103533-17 to JRY, the Helmsley Center for Genomic Medicine, the Functional Genomics Core, and the Mass Spectrometry Core of the Salk Institute with funding from NIH-NCI CCSG: P30 014195. TH holds the Renato Dulbecco Chair in Cancer Research, and is a Frank and Else Schilling American Cancer Society Professor.

## Additional information

### Competing interests

Tony Hunter: Reviewing editor, *eLife*. John R Yates III: Reviewing editor, *eLife*. The other authors declare that no competing interests exist.

### Funding

| Funder | Grant reference number | Author |
| --- | --- | --- |
| National Institute of General Medical Sciences | 8P41GM103533-17 | Aaron Aslanian<br>John R Yates III |
| National Institutes of Health | CA080100 | Zheng Wang<br>Catherine Wu<br>Aaron Aslanian<br>Tony Hunter |
| National Center for Research Resources | 5P41RR011823-17 | Aaron Aslanian<br>John R Yates III |
| National Institutes of Health | CA082683 | Zheng Wang<br>Catherine Wu<br>Aaron Aslanian<br>Tony Hunter |
| National Institutes of Health | CA014195 | Zheng Wang<br>Catherine Wu<br>Aaron Aslanian<br>Tony Hunter |

The funders had no role in study design, data collection and interpretation, or the decision to submit the work for publication.

### Author contributions
Zheng Wang, Conceptualization, Supervision, Validation, Investigation, Methodology, Writing—original draft, Project administration; Catherine Wu, Aaron Aslanian, Investigation, Project administration; John R Yates III, Conceptualization, Supervision, Funding acquisition, Writing—review and editing; Tony Hunter, Funding acquisition

### Author ORCIDs
Tony Hunter [ID] https://orcid.org/0000-0002-7691-6993

### Decision letter and Author response
Decision letter https://doi.org/10.7554/eLife.35447.032
Author response https://doi.org/10.7554/eLife.35447.033

## Additional files

### Supplementary files
• Supplementary file 1. Yeast strains used in this study.
DOI: https://doi.org/10.7554/eLife.35447.027
• Supplementary file 2. Plasmids used in this study.
DOI: https://doi.org/10.7554/eLife.35447.028
• Supplementary file 3. Primers used in this study. The same primers were used in RNA level measurement and in chromatin IP experiments. * Used as gene-specific primer in reverse transcription.
DOI: https://doi.org/10.7554/eLife.35447.029
• Transparent reporting form
DOI: https://doi.org/10.7554/eLife.35447.030

### Data availability
All data generated or analysed during this study are included in the manuscript and supporting files.

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
