## [Decision Letter]

Thank you for submitting your article "Defective RNA Polymerase III is negatively regulated by the SUMO-Ubiquitin-Cdc48 Pathway" for consideration by *eLife*. Your article has been reviewed by three peer reviewers, one of whom is a member of our Board of Reviewing Editors, and the evaluation has been overseen by Philip Cole as the Senior Editor. The following individual involved in review of your submission has agreed to reveal her identity: Deborah Johnson (Reviewer #3).

The reviewers have discussed the reviews with one another and the Reviewing Editor has drafted this decision to help you prepare a revised submission.

Summary:

Regulation of RNA polymerase III is important for eukaryotic cell division and survival, as illustrated by mutations in the RNA polymerase III machinery that have been associated with neurodegenerative diseases. Little is known about molecular mechanisms that ensure accurate transcription by RNA polymerase III. In this study, Wang and coworkers used a genetic approach to identify sumoylation as an important regulatory mechanism that inhibits RNA polymerase III dependent transcription. Their data suggests that SUMO modification of the RNA Pol III subunit Rpc53 triggers ubiquitylation of another subunit, Rpc160, which in turn leads to Cdc46-dependent disassembly of RNA polymerase III complexes. Inhibition of SUMO-modification of Rpc53 accordingly increases RNA polymerase III activity, thus allowing for genetic rescue of RNA Pol III mutants, such as those observed in human diseases. This mode of regulation was discovered by analyzing rescue of defective RNA Pol III, and thus, might be particularly important to clear stalled RNA Pol III complexes from chromosomes; indeed, this pathway operates on chromatin as shown by the authors. Overall, this is a nice study that uses a powerful combination of genetics and mass spectrometry to identify components of the SUMO-ubiquitin-Cdc48 axis that regulates RNA Pol III. The experiments have been performed well and the results are typically very clean. The findings of this study have implications on how transcription of tRNA genes is regulated in eukaryotic cells, and it might point to new approaches to treat diseases caused by mutations in the RNA Pol III machinery. However, after a discussion among the reviewers, the following three issues need to be addressed before proceeding with this manuscript in *eLife*:

Essential revisions:

1) The authors need to provide more insight into Rpc160 sumoylation and ubiquitylation. One reviewer indicated that Rpc160 might indeed by SUMO-modified, and little data is available to probe its ubiquitylation downstream of SUMO-modification. It would be particularly important to show that Rpc160 ubiquitylation is affected in strains that express the *rpc53-3KR* mutant.

2) In addition, the authors needs to show more directly how Rpc53 sumoylation and/or Rpc160 ubiquitylation recruit Cdc48, and how this leads to disassembly of stalled RNA Pol III complexes. Binding studies might be helpful, but the reviewers agreed that immunoprecipitations from cells expressing mutant Cdc48 could be very informative (i.e. does this have an effect of RNA Pol III composition?).

3) Finally, it would be important to strengthen the suggested role of this pathway in a stress response to stalled or inactive RNA Pol III, as discussed by the authors. Are there any growth phenotypes of *rpc53-3KR* cells? Alternatively, are Siz1, Slx5 or Cdc48 recruited to tRNA genes as a response to expression of defective RNA Pol III mutants? Such data would strengthen the physiological relevance of this study.

---

## [Author Response]

Essential revisions:1) The authors need to provide more insight into Rpc160 sumoylation and ubiquitylation. One reviewer indicated that Rpc160 might indeed by SUMO-modified, and little data is available to probe its ubiquitylation downstream of SUMO-modification. It would be particularly important to show that Rpc160 ubiquitylation is affected in strains that express the rpc53-3KR mutant.

We made a K-R mutation at the only consensus sumoylation site (K985R) in Rpc160, but the mutation did not rescue *rpc160-M809I*. In addition, the results in Figure 3A show that the major sumoylated proteins are smaller than 160 kDa, and so we have not obtained any evidence to support a role of Rpc160 sumoylation in this case.

We tried to directly detect Ub conjugated to Rpc160 both by IP and by immunoblotting, but could not obtain a convincing signal for Rpc160 ubiquitylation. We have tried isolating His-tagged ubiquitin under denaturing conditions in the presence of proteasome inhibitor MG132 in *pdr5Δ* yeast cells that are more permeable to the drug, but still could not see ubiquitylation of Rpc160. The prior study that identified Rpc160 ubiquitylation sites was performed in the similar way, but used an anti-diGly-Lys antibody to enrich tryptic peptides derived from ubiquitylated Lys residues followed by MS analysis. In spite of our negative results, given the fact that the ubiquitylation sites on Rpc160 were previously identified by mass spectrometry, and that mutating three of the sites can stabilize the mutant Rpc160-M809I and at least partially rescue its growth defect, we are confident that Rpc160 ubiquitylation must exist in vivo. Although we did not demonstrate it directly, we believe sumoylation and ubiquitylation function in a linear pathway, because 1) Slx5/Slx8 is a well-established STUbL E3 ligase, 2) Slx5 interacts with Rpc53 in a SUMO-dependent manner, 3) the SIMs in Slx5 are required for Pol III inhibition, and 4) deletion of SIZ1 or UBC4 both fully rescue the *rpc160-M809I* growth defect, suggesting they are not redundant with each other in Pol III regulation. As mutating the three ubiquitylation sites in Rpc160 only partially rescued *rpc160-M809I*, it is likely that additional ubiquitylation sites in Rpc160 and in other relevant proteins exist, which remain to be identified in the future.

2) In addition, the authors needs to show more directly how Rpc53 sumoylation and/or Rpc160 ubiquitylation recruit Cdc48, and how this leads to disassembly of stalled RNA Pol III complexes. Binding studies might be helpful, but the reviewers agreed that immunoprecipitations from cells expressing mutant Cdc48 could be very informative (i.e. does this have an effect of RNA Pol III composition?).

As suggested, we have designed an IP/MS experiment and a ChIP experiment to determine whether the subunit composition of the Pol III complex and the association between Rpc160 and chromatin is affected by Cdc48 activity, and the new results are now incorporated as Figure 5G and 5H, respectively. In summary, our new data suggest that the mutant Rpc160-M809I proteins are largely released from Pol III holoenzyme, as well as from tRNA genes, in a Cdc48-dependent manner, since the associations of Rpc160-M809I with other Pol III subunits and with chromatin can both be restored in *cdc48-3* mutant cells. Therefore, consistent with its segregase activity, Cdc48 may be able to disassemble the Pol III protein complex, as well as to disassociate Pol III complexes from the chromatin.

3) Finally, it would be important to strengthen the suggested role of this pathway in a stress response to stalled or inactive RNA Pol III, as discussed by the authors. Are there any growth phenotypes of rpc53-3KR cells? Alternatively, are Siz1, Slx5 or Cdc48 recruited to tRNA genes as a response to expression of defective RNA Pol III mutants? Such data would strengthen the physiological relevance of this study.

We have not been able to observe any growth phenotype for *rpc53-3KR* cells, and we think a synthetic lethal screen of *rpc53-3KR* can be very informative, but it is beyond the scope of this study. We did examine the recruitment of Siz1 to tRNA genes as a response to expression of defective Pol III mutants, and the results are incorporated as a new Figure 7B. In this experiment, we placed *rpc160-M809I* under control of the GAL1 promoter, which can be induced by growing cells in galactose media. We found that more Siz1 protein was recruited to tRNA genes upon induction of the mutant *rpc160-M809I*, compared with when additional wild type *RPC160* expression was induced. These results are consistent with the findings in our initial submission, that more Siz1, Slx5, and Cdc48 proteins are associated with tRNA genes, when the mutant *rpc160-M809I* allele is the only source of translated *rpc160* protein. The data suggest that the recruitment of Siz1 to tRNA genes might be a result of defective/stalled Pol III transcription, which is consistent with our protein quality control hypothesis.